# Host glutathione is required for *Rickettsia parkeri* cell division and intracellular survival

Han Sun ⬤, Anh Phuong Luu, Meggie Danielson & Thomas P. Burke ⬤ ✉

Intracellular bacteria rely on eukaryotic metabolites for their fitness and pathogenesis. Yet, the mechanisms of how host metabolites promote bacterial physiology and immune evasion are often unclear. Here, we employed obligate cytosolic *Rickettsia parkeri*, which parasitizes over fifty host metabolites, to investigate bacterial utilization of host glutathione (GSH). We observed that GSH depletion impaired *R. parkeri* intracellular survival. Super-resolution microscopy revealed that GSH depletion caused bacterial chaining in the host cytosol, prohibiting proper actin-based motility and cell-to-cell spread. GSH was especially critical for bacterial survival in primary macrophages, where it enabled *R. parkeri* to evade ubiquitylation and antibacterial autophagy. Cell division and survival defects were restored by supplementing N-acetylcysteine, suggesting that GSH serves as a cysteine source for *R. parkeri*. Together, these data suggest that *Rickettsia* requires GSH as a nutrient source to promote cell division, actin-based motility, evasion of antibacterial autophagy, and intracellular survival. This knowledge contributes to the expanding paradigm that GSH plays diverse roles in the pathogenesis of intracellular bacteria and represents a potential target for host-acting therapeutics.

Spotted fever group (SFG) *Rickettsia* species are dangerous agents of disease worldwide[1]. SFG *R. rickettsii* is the causative agent of Rocky Mountain spotted fever (RMSF), a disease characterized by fever of over 104° F, maculopapular rash, severe headache, and in some cases death[2]. Case-fatality rates for RMSF were as high as 65–80% prior to antibiotics[3] and can be fatal even after antibiotic treatment[4]. The SFG pathogen *R. parkeri* is the causative agent of mild spotted fever disease in North and South America and has a highly similar genome to *R. rickettsii*, with some virulence genes containing >99% sequence alignment, with the major difference being a single 33.5 kb locus that is absent in *R. rickettsii*[5–7]. *R. parkeri* rickettsiosis is characterized by a skin lesion (eschar) at the site of tick bite, fever of 103° F, rash, and headache[6,7]. Spotted fever has increased in incidence from approximately 500 cases in the year 2000 to 6248 cases in 2017, and disease occurs in every state of the continental United States[1]. Spotted fever and other tick-borne diseases are increasing in prevalence as the range of ticks expands, which is likely impacted by climate change[8]. The increasing incidence combined with difficulties in recognizing spotted fever clinically[9] illustrate a critical need to better understand molecular mechanisms of disease caused by SFG rickettsial pathogens.

*Rickettsia* and their close relatives are obligate to the host cell cytosol, where they parasitize over fifty metabolites[10,11]. *Rickettsia* have highly reduced genomes with degraded biosynthetic pathways, including the loss of enzymes required for glycolysis, as well as biosynthesis of fatty acids, isoprenoids, vitamins, cofactors and amino acids, and the abundant low molecular weight thiol glutathione (GSH)[5,10,12]. GSH is a tripeptide antioxidant produced in nearly all ecosystems, and is the most abundant low molecular weight thiol in plant and mammalian cells[13,14]. It protects cells from oxidative damage and helps maintain redox homeostasis, for example by removing peroxides, through its reactive sulfhydryl group (-SH)[13]. S-glutathionylation of cysteine is a post-translational modification that alters protein activity[13]. Thus, GSH can serve three main purposes: as a source of cysteine, maintaining redox homeostasis and protecting from oxidative damage, and as a post-translational modification that alters protein activity.

Department of Microbiology and Molecular Genetics, School of Medicine, University of California, Irvine, Irvine, CA 92617, USA. ✉ e-mail: tpburke@uci.edu

GSH is acquired, sensed, and produced by phylogenetically distinct cytosol-dwelling pathogens, including *Listeria monocytogenes, Burkholderia pseudomallei* and *Francisella* species. An elegant example is in *B. pseudomallei*, whereby an extracellular histidine kinase domain binds GSH to initiate expression of the type VI secretion system to initiate virulence[15]. Inhibition of GSH synthesis by buthionine sulfoximine (BSO), which is a potent, specific, non-toxic, and irreversible inhibitor of host γ-glutamylcysteine synthetase, blocks *B. pseudomallei* virulence gene expression and inhibits cell to cell spread[15]. In another striking example of the critical role GSH plays in virulence, *L. monocytogenes* senses reducing environments and then produces its own GSH that allosterically binds and activates the master virulence gene regulator PrfA[16–18]. Exogenous reducing agents as well as methylglyoxal can also activate PrfA in liquid media, and *L. monocytogenes* can import GSH to activate virulence genes[16,18,19]. In *Francisella* species, GSH is imported and degraded as a sulfur source[20–22]. Other pathogens including *Streptococcus* species and *Hemophilus influenzae* co-opt host GSH to defend against oxidative stress[23,24]. These studies suggest that GSH plays important and varied roles in physiology and virulence of intracellular bacteria.

Here, we investigated the role for GSH in the intracellular growth and survival of the obligate cytosolic pathogen *R. parkeri*. We find that host GSH is required for intracellular survival, and most strikingly, GSH depletion causes intracellular chaining of *R. parkeri*. GSH depletion led to defects in actin-based motility and cell-to-cell spread. We also observed that GSH was more important for *R. parkeri* survival in macrophages than in epithelial or endothelial cells, whereby antibacterial autophagy was partially responsible for restricting *R. parkeri* upon GSH depletion. Mechanistically, cell division and survival were restored upon the addition of an alternative cysteine source, suggesting that GSH is a key cysteine source that promotes *Rickettsia* cell division. These findings establish a critical role for GSH in promoting *Rickettsia* cell division and pathogenesis, contributing to the concept that GSH plays multifaceted roles in the fitness of intracellular pathogens. These findings have important implications on developing host-targeted therapeutics as a novel antibacterial strategy.

## Results

### GSH is required for cell division and survival of *R. parkeri* in epithelial cells

We hypothesized that GSH was important for *R. parkeri* intracellular survival. To test this, plaque forming units (PFUs) were measured over time upon infection of Vero epithelial cells treated with BSO. GSH depletion significantly reduced the number of recoverable bacteria (Fig. 1a; all *P*-values are reported in Supplementary Table 1 and source data are available in the Dryad repository doi:10.5061/dryad.905qfttvz). BSO reduces GSH pools to below 1% in many cell types[25,26] and is generally well tolerated, including in studies on other intracellular bacteria where it was used at millimolar concentrations[15,17]. Yet, toxicity can occur depending on the cell type and culture conditions[26–29]. BSO has also been used in human clinical trials for cancer, where it caused low grade toxicity[30], suggesting it is not highly cytotoxic. To determine if BSO caused toxicity to host cells in our system, cell viability was measured throughout the infections. BSO-treated cells had a normal appearance with no apparent cell death, as observed by brightfield microscopy (Fig. 1b). A resazurin-based cell death assay also found no significant increase in cell death upon BSO treatment (Fig. 1c). These data demonstrate that *R. parkeri* requires GSH for growth and survival in epithelial cells.

To better understand how GSH promoted *R. parkeri* intracellular survival, we observed infection with immunofluorescence microscopy using a Zeiss 900 with Airyscan 2 detector that enables super-resolution imaging. Interestingly, at 48 h postinfection (hpi) *R. parkeri* in BSO-treated Vero cells were elongated into chained rods, some of which measured over 10 μM in length (Fig. 1d). When stained with an α-

*Rickettsia* antibody, the bacteria appeared either filamentous (Fig. 1d), or as individual rods within a chain (Fig. 1e). Bacterial filamentation can be elicited by a variety of stressors including DNA damage, whereby the widely conserved stress response protein RecA inhibits cell division, causing bacteria to filament into long chains with a shared cytosol[31]. To determine if the bacteria were filamentous or individual rods within a chain, we infected cells with *R. parkeri* expressing the GFP derivative AausFP1[32,33], which is expressed only in the bacterial cytosol and thus would discern compartmentalization of individual cytosols. Vero cells infected with *AausFP1*-expressing *R. parkeri* were fixed and stained with a *Rickettsia*-specific antibody. Whereas the antibody signal in certain bacteria appeared continuous across the elongated bacteria (Fig. 1f, red), the GFP signal from the GFP expressed in the bacterial cytosol was segmented in 100% of the bacteria (Fig. 1f, green). This demonstrated that the cell division defect was due to an inability of the bacteria to properly separate at their poles and that they did not have a shared cytosol, suggesting that the chaining was via a RecA-independent mechanism. Together, these findings demonstrate that host GSH is required for proper cell division and intracellular survival of *R. parkeri*.

To determine the kinetics of chaining, a time course of infection was performed from 24–96 hpi. *R. parkeri* had normal morphology in early during infection (≤24 h) in BSO-treated cells, but the bacteria were observed in chains of 2, 3, 4, and >4 bacteria from 48 hpi and beyond (Fig. 1g, quantified in **1h**). Accordingly, bacterial length increased over time in BSO-treated cells (Fig. 1i), suggesting that the bacteria were growing but unable to divide into individual ~1 μm bacteria with normal morphology. Quantifying the number of bacteria per Vero cell using microscopy revealed that *R. parkeri* abundance moderately increased from 24–72 hpi in BSO-treated cells, which was reduced as compared to untreated cells (Fig. 1j). Comparing the decreasing number of recoverable bacteria by PFU assay (Fig. 1a) versus the moderately increasing number of bacteria by microscopy (Fig. 1j) suggests that the chained bacteria are defective for their ability to invade new cells upon host cell lysis. Together, these data demonstrate that GSH is critical for *R. parkeri* cell division, intracellular survival, and infectivity.

### GSH is required for proper actin-based motility and cell-to-cell spread of *R. parkeri*

We hypothesized that the aberrant morphology of *R. parkeri* upon GSH depletion would alter the ability of the bacteria to undergo proper actin-based motility, which promotes cell-to-cell spread. *R. parkeri* undergoes two temporally distinct phases of actin-based motility: an early phase mediated by RickA that results in short, curly actin tails, and a later phase mediated by Sca2 that results in long tails and is the major mediator of cell-to-cell spread[34–37]. The frequency of actin-tail formation and actin colocalization was quantified, revealing that untreated cells had high frequencies of actin tails at 48 hpi (Fig. 2a). In contrast, depleting GSH reduced the frequency of long actin tails at 48 h as compared to untreated cells (Fig. 2a). Quantifying actin tails over time from 30 mpi to 72 hpi revealed that GSH promoted actin tail formation (Fig. 2a). Interestingly, filamentous actin was often observed at the bacterial poles within chains of bacteria, providing further evidence that the bacteria do not have a shared cytosol (Fig. 2a, long white arrows). As many *Rickettsia* species target endothelial cells during systemic infection[38], we determined the effects of BSO on infected human microvascular endothelial cells (HMEC-1). Similar phenotypes were observed between HMEC-1 cells and Vero cells, including defects in cell division and actin-based motility upon GSH depletion (Fig. 2c). We hypothesized that the defects in cell division and actin-based motility would hamper *R. parkeri* cell-to-cell spread and plaque formation. Vero cells depleted for GSH were infected with *R. parkeri* and plaque formation was measured at 7 days

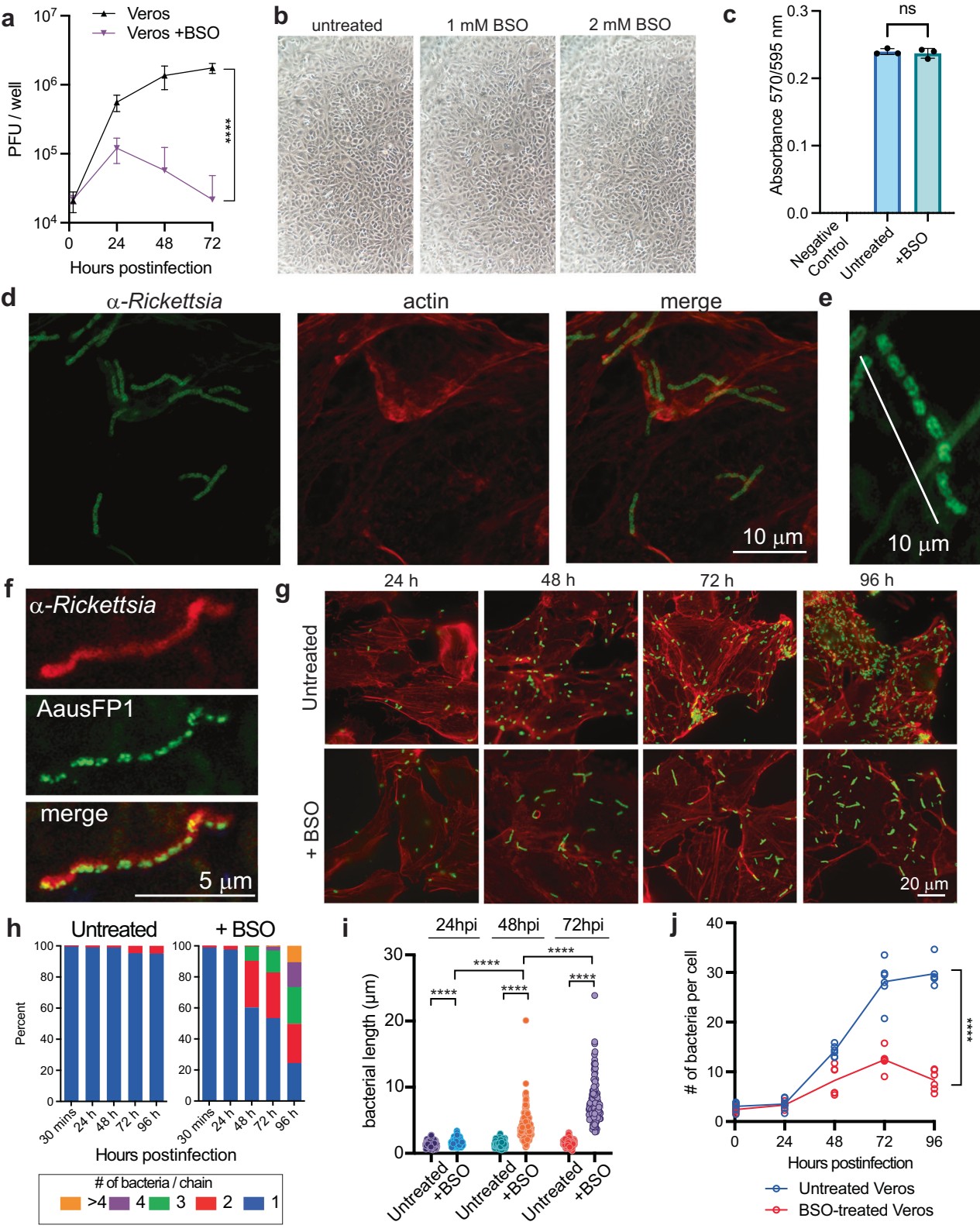

postinfection (dpi). In parallel, to determine if BSO acted directly on the bacteria themselves, we incubated *R. parkeri* with BSO immediately prior to infection of untreated Vero cells. Strikingly, whereas treatment of *R. parkeri* with BSO did not alter plaque formation, *R. parkeri* was completely unable to form distinct plaques in BSO-treated Vero cells (Fig. 2d). These data demonstrate that host GSH is required for proper actin-based motility and spread of *R. parkeri*.

## GSH is especially critical for *R. parkeri* survival in primary macrophages

Many spotted fever group *Rickettsia* associate with leukocytes throughout their infectious lifecycles, including macrophages and neutrophils in the skin and in internal organs upon dissemination[7,39–42]. We therefore sought to determine if GSH was required for *R. parkeri* survival in primary macrophages. In murine bone marrow-derived macrophages (BMDMs), depletion of GSH with BSO led to a reduction

**Fig. 1 | GSH is required for cell division and growth of _R. parkeri_ in epithelial cells. a** _R. parkeri_ abundance in Vero cells infected at an MOI of 1, untreated or treated with 2 mM BSO overnight prior to infection. Infected cells were lysed at the indicated times and plated for PFU. Data are the compilation of four separate experiments and are expressed as means ± SD. Statistics used a two-way ANOVA. **b** Brightfield images (10x) of Vero cells treated with the indicated concentrations of BSO, 48 h after treatment. **c** Cell death assays using resazurin viability assay. Absorbance is 570/595 nm. Each point is an independent experiment. Statistics used a one-way ANOVA, ns not significant. **d, e** Representative images of 63x confocal super-resolution microscopy of Vero cells treated with 2 mM BSO and infected with _R. parkeri_, at 48 hpi. Coverslips were stained with phalloidin to label actin (red) and with an α-_Rickettsia_ antibody (green). **f** Representative images of 63x confocal super-resolution microscopy of Vero cells treated with 2 mM BSO and infected with AausFP1-expressing _R. parkeri_ (green), at 48 hpi. Coverslips were stained with α-_Rickettsia_ antibody (red). Data are representative of at least three separate experiments. **g** Representative immunofluorescence microscopy images of Vero cells untreated or treated with 2 mM BSO and infected with _R. parkeri_. Coverslips were stained with phalloidin to label actin (red) and with an anti-_Rickettsia_ antibody (green). **h** Quantification of images from untreated Vero cells (left) or Vero cells treated with 2 mM BSO overnight (right), from images in (**g**). **i** Quantification of bacterial length in Vero cells over time, from images in (**g**). BSO was added overnight at 2 mM. $n = 170$ for all data sets except +BSO treated at 72 hpi, $n = 112$. Data were compared with a two-way Student's _T_-test. **j** Quantification of data from (**g**). Data are the compilation of 6 separate experiments and are expressed as means ± SD. Statistics were performed with a two-way ANOVA, ****$p < 0.0001$.

in recoverable PFUs that was dramatically more severe than in epithelial cells and led to complete restriction of the bacteria by ~48 hpi (Fig. 3a). As macrophages have enhanced antimicrobial killing capabilities, we hypothesized that the bacteria were being killed. To determine whether the loss of recoverable PFUs was due to bacteria being killed or due to the presence of chained bacteria that cannot form plaques (as we observed in epithelial cells), we analyzed infected BMDMs with immunofluorescence microscopy. This revealed that _R. parkeri_ was largely absent from BMDMs at 48 hpi (Fig. 3b), suggesting that _R. parkeri_ were being killed and removed from macrophages.

The explanation for this striking clearance of _R. parkeri_ from macrophages was unclear, and we hypothesized that GSH was either promoting the ability of _R. parkeri_ to evade inflammasomes or antibacterial autophagy. We previously described how lysis of _R. parkeri_ in macrophages by guanylate binding proteins (GBPs) causes caspase 11 inflammasome activation and pyroptosis[43]. In the absence of the inflammasome, bacteriolysis instead activates the DNA sensor cGAS, causing type I interferon (IFN-I) production[43]. We therefore used cell death and IFN-I production as proxies to determine if GSH depletion caused bacteriolysis. Supernatants from infected BMDMs were collected to measure IFN-I production and cell death, which was measured using a lactate dehydrogenase (LDH) release assay for pyroptosis. We measured host cell death in WT macrophages and IFN-I release in _Casp1/11-/-_ macrophages in the presence or absence of BSO. In WT macrophages, BSO treatment resulted in similar LDH release as in untreated cells (Fig. 3c), suggesting that GSH was not required for protecting against bacteriolysis. Infection of BSO-treated and untreated BMDMs resulted in similar amounts of IFN-I production in WT cells. IFN-I was reduced in BSO-treated _Casp1/11-/-_ cells as compared to untreated cells (Fig. 3d). The reduction in IFN-I may be due to a reduced number of bacteria in these cells, which is complicated by the fact that _R. parkeri_ suffers in _Casp1/11-/-_ due to increased IFN-I[43], but nevertheless there was no increase in IFN-I that would suggest an increase in bacteriolysis. This suggested that _R. parkeri_ killed by GSH depletion were not hyper-activating the inflammasome.

To determine if _R. parkeri_ restriction by BSO activated other innate immune pathways, we also measured NF-κB activation and found that while _R. parkeri_ elicited higher NF-κB activity as compared to uninfected cells, BSO treatment did not alter this response (Fig. 3e). These data align with the LDH and IFN-I data suggesting that GSH depletion does not cause significant bacteriolysis that hyper-activates innate immune pathways. Together, these data demonstrate that GSH is especially critical for _R. parkeri_ survival in macrophages as compared to epithelial cells and protects them from a non-bacteriolytic, inflammasome-independent mechanism of restriction.

**GSH enables _R. parkeri_ to avoid ubiquitin targeting and restriction by antibacterial autophagy**

_R. parkeri_ evolved multiple mechanisms to evade ubiquitylation[44–46], and since restriction of _R. parkeri_ by BSO in macrophages did not result in bacteriolysis (Fig. 3), we next hypothesized that chained bacteria were being targeted by ubiquitylation and antibacterial autophagy. We therefore examined whether GSH depletion increased polyubiquitin recruitment to _R. parkeri_. Immunofluorescence microscopy revealed that GSH depletion significantly increased labeling of _R. parkeri_ by polyubiquitin in Vero cells (Fig. 4a, b). We then sought to determine whether increased ubiquitin targeting led to restriction by antibacterial autophagy. Previous studies found that _R. parkeri_ mutants unable to avoid targeting by ubiquitin are restricted in BMDMs in a process that requires the autophagy receptor ATG5[44,45]. We therefore examined whether GSH depletion restricted _R. parkeri_ in an ATG5-dependent manner. Indeed, there was a 44-fold increase in bacterial burdens in _LysM-Atg5-/-_ BMDMs as compared to WT BMDMs or control _Atg5fl/fl_ BMDMs upon BSO treatment (Fig. 4c), suggesting that autophagy was partially responsible for killing _R. parkeri_ in the absence of GSH. As ATG5 has other roles outside of antibacterial autophagy, we also asked whether inhibiting autophagy with a small molecule (3-methyladenine, 3MA) altered _R. parkeri_ abundance in the presence of BSO. 3MA treatment increased _R. parkeri_ abundance 34-fold in cells depleted for GSH (Fig. 4d), similar to _Atg5_ deletion. Inhibiting autophagy via _Atg5_ deletion or 3MA treatment did not fully restore _R. parkeri_ growth in GSH-depleted cells, and therefore we conclude that autophagy-mediated restriction plays an important but not complete role in killing the bacteria. As autophagy mediates a non-bacteriolytic mechanism of restriction, these findings align with the findings on an inflammasome-independent mechanism of restriction and demonstrate that host GSH is required for _R. parkeri_ to avoid ubiquitylation and antibacterial autophagy in macrophages.

**Exogenous N-acetylcysteine restores _R. parkeri_ division and survival in the absence of GSH**

The mechanisms regarding how GSH contributes to _R. parkeri_ division and intracellular survival were unclear, as GSH can detoxify ROS, modify proteins post-translationally, and serve as a cysteine/sulfur nutrient source. To discriminate between these possibilities, we supplemented the media of BSO-treated cells with different cysteine-containing metabolites. N-acetylcysteine (NAC) is widely regarded and used as a prodrug to increase intracellular cysteine pools and is used to treat glutathione deficiency diseases[47]. We also sought to test other cysteine sources, however there are challenges with solubility and import of other cysteine-containing molecules. For example, GSH itself is not imported, but rather GSH ethyl ester (GSHee) is imported[25]. Second, cystine is the primary imported form of cysteine[27], but both cystine and cysteine are challenging to use due to rapid oxidation and insolubility[48]. Moreover, cystine transporters often are glutamate antiporters[49], which can limit cystine import. Third, the dipeptide cysteinylglycine (CysGly) which is an intermediate breakdown product of GSH was tested although its relative import rate as compared to other cysteine-containing derivatives is unknown. Notably, under BSO treatment, these molecules may increase cysteine availability but should not restore GSH levels, as BSO irreversibly inhibits

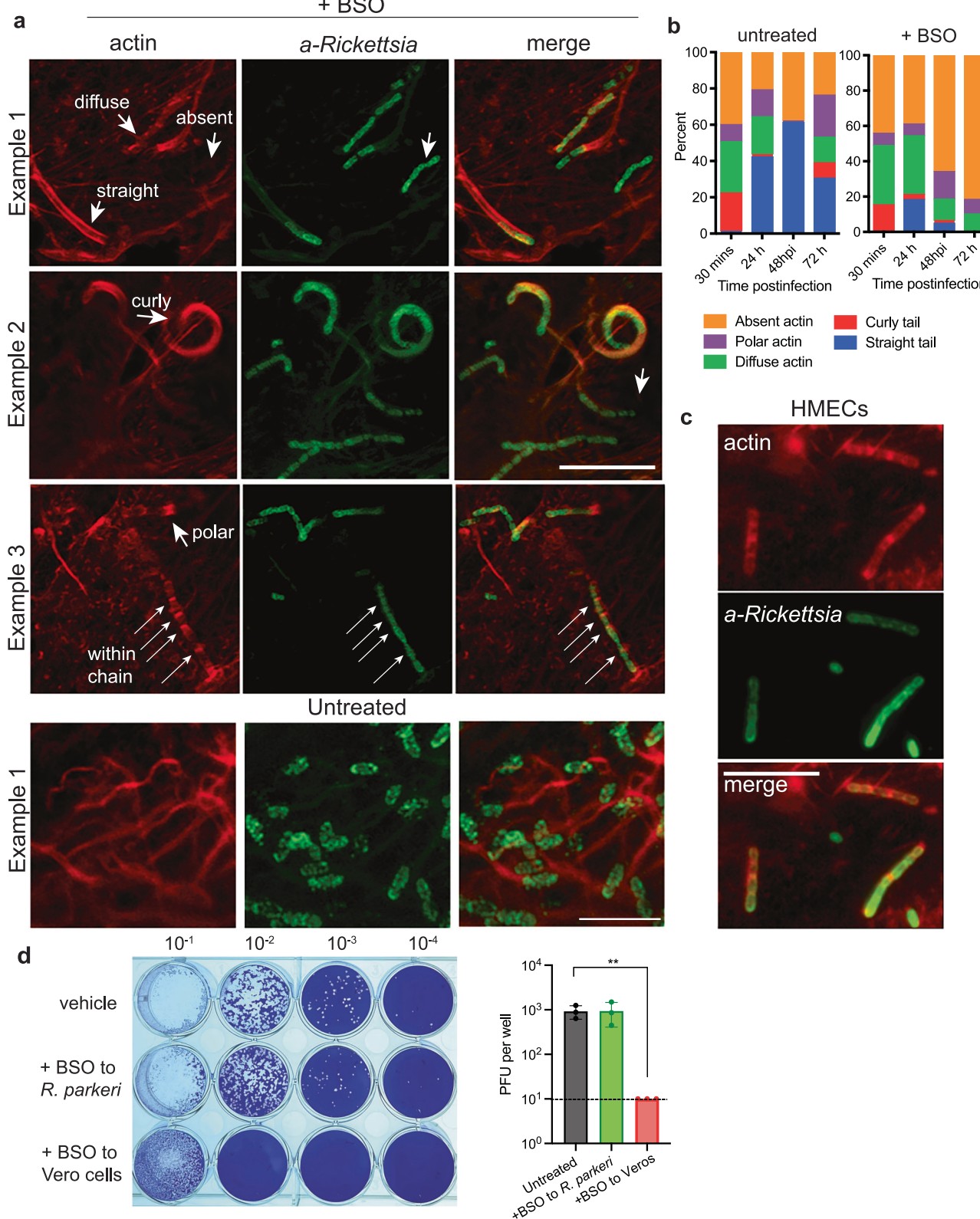

γ-glutamylcysteine synthetase (γ-GCS) the enzyme responsible for GSH synthesis downstream of cysteine[48].

Upon the exogenous addition of GSHee to BSO-treated BMDMs, *R. parkeri* survival was significantly increased (Fig. 5a), and cell division was restored in Vero cells (Fig. 5b), suggesting that GSH is indeed responsible for the observed phenotypes upon γ-GCS inhibition by BSO. Interestingly, media supplemented with NAC also significantly

restored survival in BMDMs (Fig. 5a) and cell division in Vero cells (Fig. 5b). Neither GSH itself, CysGly, nor cystine restored division or survival (Supplemental Fig. 1). The explanation for why NAC and GSHee restored *R. parkeri* division and survival while the other cysteine-containing molecules did not is unclear but could likely be due to import rates, as NAC is better taken up by cells, less prone to oxidation, and for these reasons is widely used both in tissue culture

**Fig. 2 | Host GSH is required for proper actin-based motility and cell-to-cell spread of *R. parkeri*. a** Representative images of Vero cells infected with *R. parkeri* at an MOI of 1 and imaged at 48 hpi. Coverslips were stained with phalloidin to label actin (red) and with an α-*Rickettsia* antibody (green). 2 mM BSO was added prior to infection overnight prior to infection. Scale bar = 10 μm in BSO treated images. Scale bar = 5 μm in untreated images. **b** Quantification of actin colocalization on *R. parkeri*. Each time point is an average of at least 4 separate experiments totaling > 400 bacteria. **c** Representative images of HMEC-1 cells treated with 2 mM BSO overnight and infected with *R. parkeri*, at 48 hpi. Coverslips were stained with phalloidin to label actin (red) and with an α-*Rickettsia* antibody (green). Scale

bar = 5 μm. Images in (**a**) were captured with 63x confocal immunofluorescence microscopy. Images in (**c**) were captured with 100x confocal microscopy. Images represent 3 independent experiments. **d** Vero cells were infected with *R. parkeri* at the indicated dilutions. For treatment of Vero cells, 2 mM BSO was added 16 h prior to infection, removed prior to infection, and again added at 30 mpi. For treatment of *R. parkeri* with BSO, bacteria were incubated on ice with 2 mM BSO for 15 m prior to infection. Image is representative of three independent experiments, as shown on the right, in which the second dilutions are quantified. Data were compared with a two-way Student's *T*-test. Data are expressed as means ± SD.

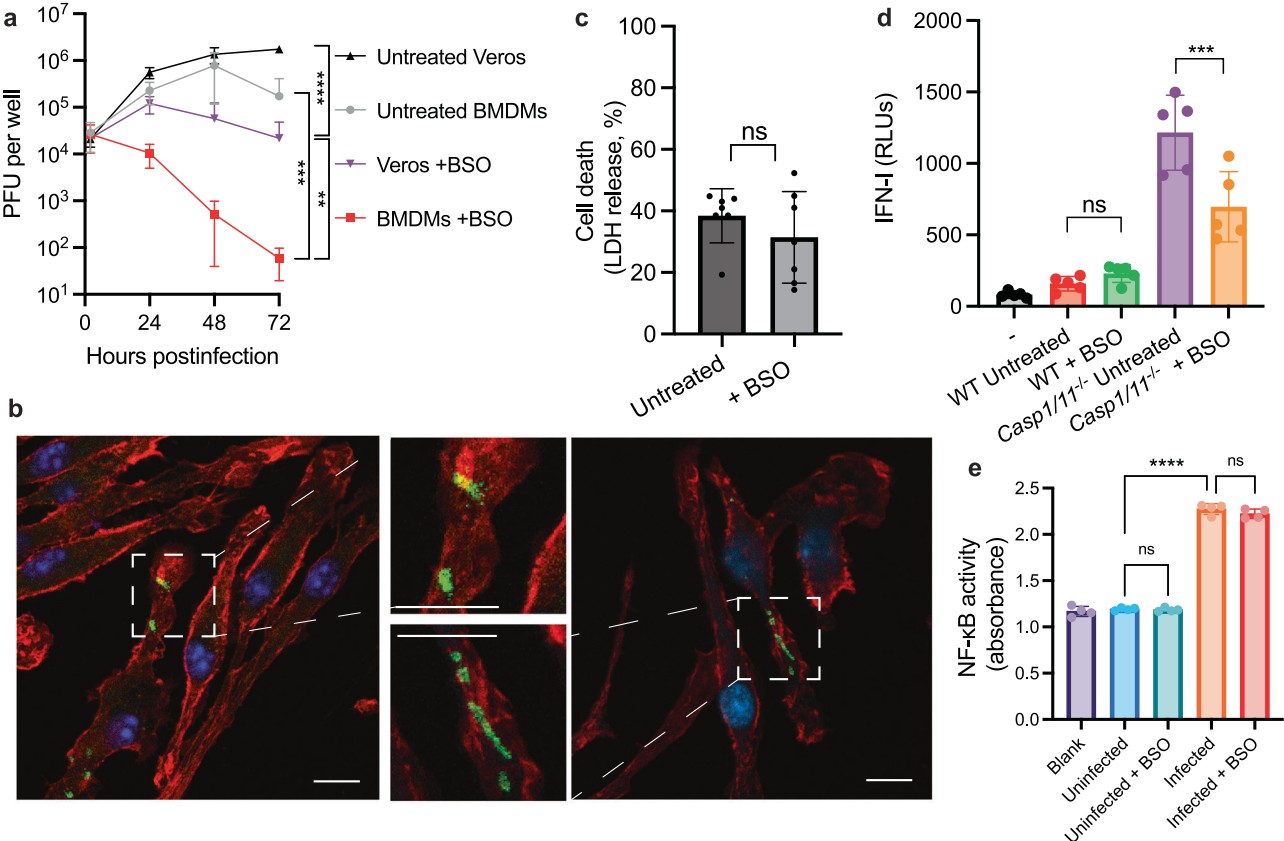

**Fig. 3 | Host GSH is critical for *R. parkeri* survival in macrophages via a non-bacteriolytic restriction mechanism. a** *R. parkeri* abundance WT murine BMDMs or Vero cells infected at an MOI of 1. Data are the compilation of six separate experiments for BMDMs and four separate experiments for Veros and are expressed as means ± SD. Statistics used a two-way ANOVA. **b** Representative images using 63x confocal microscopy of BMDMs treated with 2 mM BSO overnight, infected with *R. parkeri* at an MOI of 1 and imaged at 48 hpi. Green = α-*Rickettsia* antibody; red = phalloidin (actin), blue = DAPI. Scale bars = 10 μm. **c** Quantification of host cell death during *R. parkeri* infection of WT BMDMs. LDH release was measured at 24 hpi upon *R. parkeri* infection at an MOI of 1. Data are the compilation of seven separate experiments and are expressed as means ± SD.

Statistics used a two-way Student's *T*-test. **d** Measurement of IFN-I in supernatants of BMDMs infected with *R. parkeri* at an MOI of 1. Supernatants were harvested at 24 hpi and used to stimulate a luciferase-expressing cell line (ISRE) and relative light units (RLU) were measured and compared between treated and untreated cells. Data are the compilation of five separate experiments and are expressed as means ± SD. Statistics used a one-way ANOVA. **e** RAW-blue NF-κB reporter cells were infected with *R. parkeri* at an MOI of 1 for 48 h and supernatants were then incubated with Quantiblue for 2 h prior to measuring absorbance. Absorbance was read at 620 nm. Statistics used a one-way ANOVA. Data are the compilation of four separate experiments and are expressed as means ± SD. **p < 0.01, ***p < 0.001, ****p < 0.0001, ns not significant.

and in people as an antioxidant[47,48]. It could also be possible that BSO has an additional unknown activity, for example inhibiting cystine uptake. Nevertheless, these data suggest that the effects of GSH on *R. parkeri* division and survival are likely not via glutathionylation, which requires GSH itself and would not be restored by NAC. Instead, these data suggest that the role for GSH on *R. parkeri* is either as a cysteine source or by neutralizing ROS.

To discriminate between the role for GSH in either resisting ROS or as a nutrient source, we sought to determine if depleting GSH increased ROS. Previous studies found increased ROS at 48 h post BSO treatment in a B cell line[50,51]. In alignment with this, we found that BSO slightly (1.5-fold) but insignificantly increased ROS in Vero cells

(Fig. 5c). To determine if increased ROS was responsible for the chaining and restriction in BMDMs, we treated cells with 20 μM hydrogen peroxide ($H_2O_2$). This concentration was reported to elicit little toxicity on cells[52], and throughout the course of infection, we did not observe significant cell death after $H_2O_2$ treatment. To confirm that $H_2O_2$ increased ROS, we used DCFH-DA to measure ROS and observed that it was significantly increased 4 h after treatment (Fig. 5c). $H_2O_2$ added at 1, 24, and 48 hpi to infected BMDMs did not restrict *R. parkeri* survival in BMDMs (Fig. 5d). Similarly, *R. parkeri* in Vero cells treated with $H_2O_2$ at 1 and 24 hpi had similar morphology as in untreated cells (Fig. 5e). *R. parkeri* infection itself did not increase ROS production and may have decreased ROS (p = 0.0579, Fig. 5f). These data demonstrate

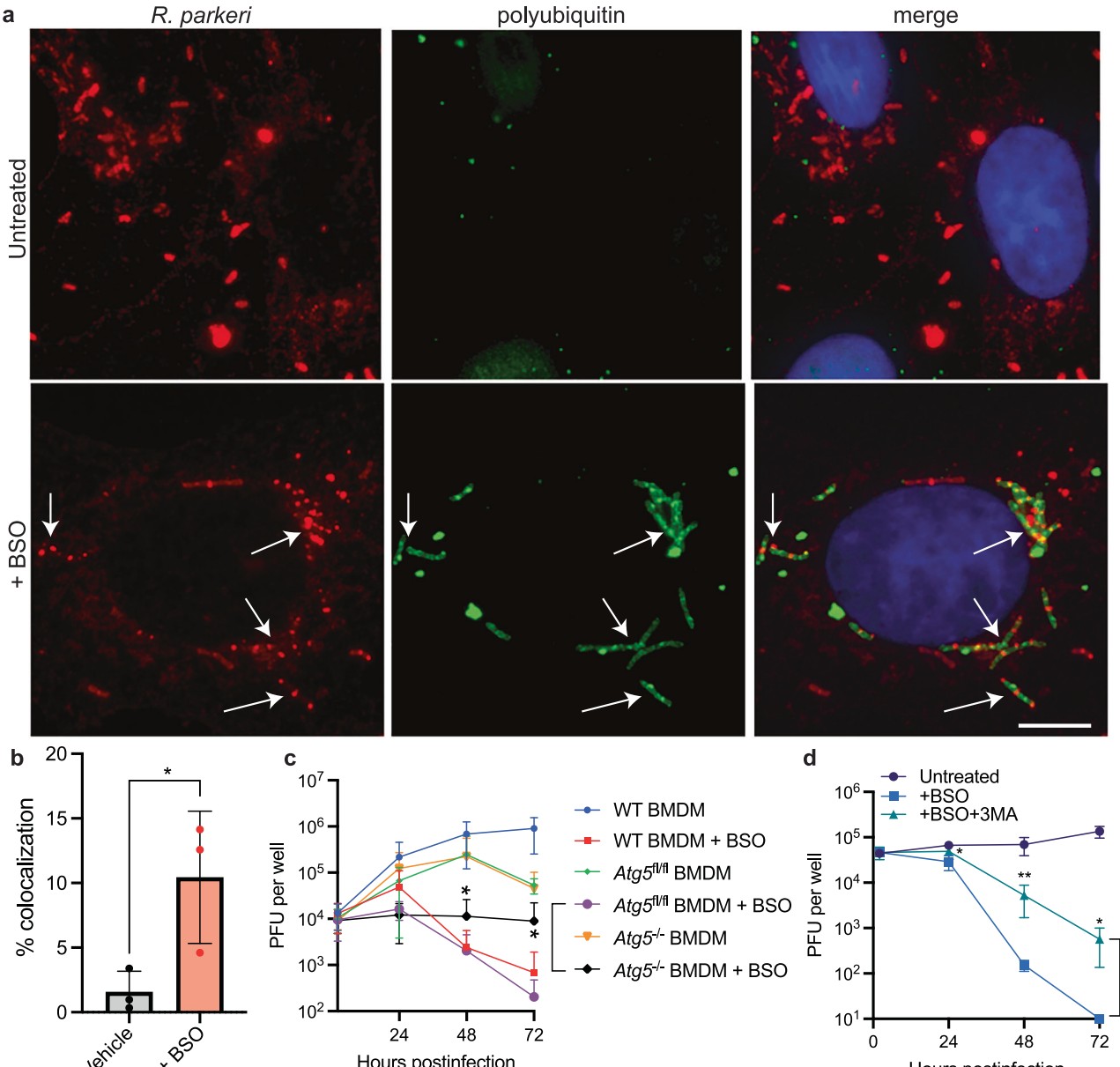

**Fig. 4 | Host GSH is required for *R. parkeri* to avoid ubiquitylation and antibacterial autophagy in macrophages. a** Representative images of ubiquitin colocalization using 20x immunofluorescence microscopy of Vero cells infected with *R. parkeri* at an MOI of 1 and fixed at 48 hpi. Cells were stained with FK1 α-polyubiquitin (green), α-*Rickettsia* I7205 (red); and DAPI (blue). Scale bar = 10 μm. **b** Quantification of ubiquitin colocalization at 48 hpi in Vero cells. Each data point is an independent experiment where at least 3 images were analyzed, and each image contained at least 50 bacteria. Statistics used a two-way Student's *T*-test, and data are expressed as means ± SD. **c** *R. parkeri* abundance in BMDMs infected at an MOI of 1. Cells treated with 2 mM BSO were incubated overnight with BSO. Data are the compilation of six separate experiments and are expressed as means ± SD. Statistics used a two-way Student's *T*-test. **d** *R. parkeri* abundance in BMDMs that were untreated or treated with 3MA and infected at an MOI of 1. Cells treated with 2 mM BSO were incubated overnight and 3MA was used at 5 mM added at 1 hpi. Data are the compilation of six separate experiments and are expressed as means ± SD. Statistical analysis used a two-way Student's *T*-test; *$p < 0.05$, **$p < 0.01$.

that the requirement for GSH in *R. parkeri* cell division and survival is not mediated through detoxifying ROS, and we therefore conclude that GSH likely functions as a nutrient source for cysteine.

To further test a potential role for GSH in protecting *R. parkeri* from ROS, we hypothesized that if the antimicrobial effects of BSO were mediated through ROS, bacterial abundance would be increased upon inhibiting nitric oxide (NO) production. However, the inducible NO synthase (iNOS) inhibitor L-NIL did not increase bacterial burdens in BSO-treated WT BMDMs (Fig. 5g). We also asked if the antimicrobial effects of NO were more apparent in the absence of antibacterial autophagy; however, iNOS inhibition did not alter bacterial burdens in BSO-treated *Atg5*⁻/⁻ BMDMs (Supplemental Fig. 2). This suggested that

antibacterial autophagy was not masking any potential antimicrobial effects of ROS. Together, these data suggest that increased ROS does not elicit chaining or bacterial killing, and we conclude that GSH likely contributes to *R. parkeri* cell division and intracellular survival by serving as a source of cysteine.

## Discussion

Intracellular bacterial pathogens must rely on eukaryotic metabolites for their fitness and pathogenesis. As the only obligate cytosolic pathogens, *Rickettsia* and their close relatives are useful models to understand how bacteria parasitize eukaryotic metabolites, as these microbes steal over fifty nutrients from their hosts and evolved a

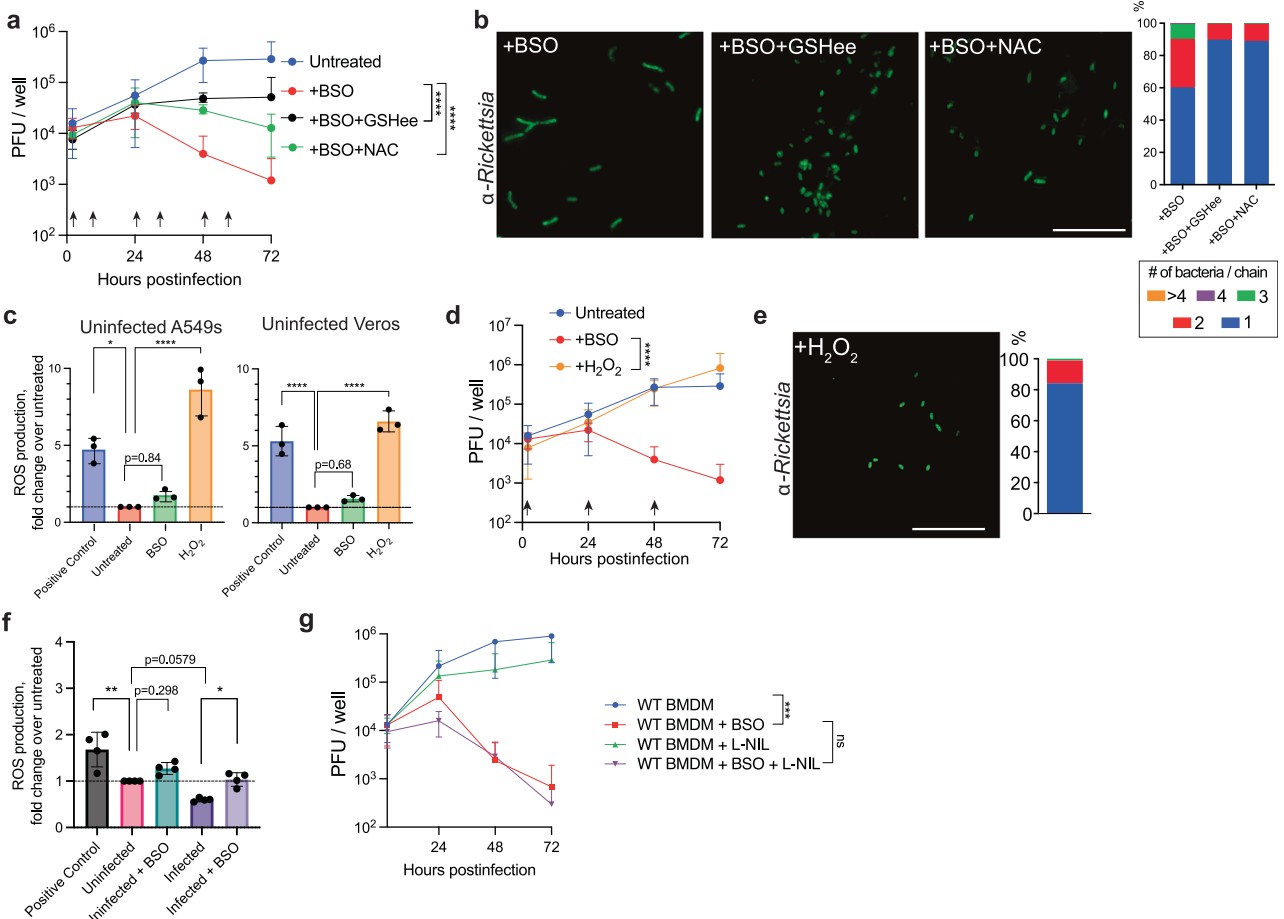

**Fig. 5 | GSH serves as a cysteine source for *R. parkeri* cell division and survival.**
**a** PFUs of *R. parkeri* in BMDMs upon addition of metabolites. 5 mM GSHee and 5 mM NAC were added when indicated by the black arrows. Data are combined from six separate experiments and are means ± SD. Statistics used a two-way Student's *T*-test at 72 hpi. **b** Representative images and quantification (right) of *R. parkeri* in Vero cells at 48 hpi in the presence of the indicated metabolites. Coverslips were stained with α-*Rickettsia* antibody (green). Scale bar = 10 μm. 3 separate experiments were counted, using 3 images per experiment and > 200 bacteria total. **c** Measurements of ROS using DCFH-DA in the indicated cells. BSO (2 mM) was added overnight prior to measurements and H₂O₂ (20 μM) was added 4 h prior to measurement. ROS production fold change is compared to untreated. Data combined three separate experiments and are expressed as means ± SD. Statistics used a one-way ANOVA. **d** PFUs in BMDMs in the presence of 20 μM H₂O₂ added at times indicated by the black arrows. Data are the compilation of six separate

experiments and are expressed as means ± SD. Statistics used a two-way Student's *T*-test at 72 hpi. **e** Representative image and quantification (right) of *R. parkeri* in Vero cells at 48 hpi in the presence of 20 μM H₂O₂. Coverslips were stained with an α-*Rickettsia* antibody (green). Scale bar = 10 μm. 3 separate experiments were counted, using 3 images per experiment and > 200 bacteria total.
**f** Measurements of ROS using DCFH-DA in the indicated cells. BSO (2 mM) was added overnight prior to measurements and H₂O₂ (20 μM) was added 4 h prior to measurement. ROS production fold change is compared to untreated. Data are the compilation of four separate experiments and are expressed as means ± SD. Statistics used a one-way ANOVA. **g** *R. parkeri* PFUs in BMDMs in the presence of the indicated inhibitors; L-NIL (1 mM) was added 1 hpi; BSO (2 mM) was added overnight prior to infection. Data are the compilation of six separate experiments and are expressed as means ± SD. Statistics used a two-way ANOVA. *$p < 0.05$, **$p < 0.01$, ***$p < 0.001$, ****$p < 0.0001$.

---

reduced genome that lost the ability to synthesize many metabolites de novo, including GSH[10]. However, the roles for many host metabolites including GSH in *R. parkeri* fitness, pathogenesis and innate immune evasion are unclear. In this study we found that host GSH was required for *R. parkeri* cell division, proper actin-based motility, avoiding antibacterial autophagy, and intracellular survival. The data support the conclusion that GSH likely promotes *R. parkeri* physiology and pathogenesis by serving as a cysteine source. This knowledge contributes to the growing paradigm that GSH is critical for diverse aspects of pathogenesis and physiology of intracellular bacteria, suggesting a strategy for targeting host metabolism as a therapeutic for potentially many pathogens.

GSH plays a multitude of roles in the virulence and physiology of phylogenetically distinct pathogens. Related to physiology, GSH is imported and degraded as a sulfur source in *Francisella* species[20–22], while *Streptococcus* species and *Hemophilus influenzae* rely on host GSH to defend against oxidative stress[23,24]. Related to virulence gene

expression, GSH reduces the *B. pseudomallei* histidine kinase VirA, breaking its dimer and resulting in transcription of the type VI secretion system[15]. *L. monocytogenes* has perhaps the most complex relationship with GSH that has been described, as it not only imports GSH as a cysteine source with a dedicated GSH transporter[19], but it senses the host reducing environment and synthesizes its own GSH that post-translationally modifies and activates the master virulence transcription factor PrfA[16–18]. The addition of exogenous reducing agents including GSH to liquid media also activates PrfA[16]. Thus, *L. monocytogenes* uses GSH as a cysteine source, it senses redox, and it produces its own GSH to initiate virulence. Our findings reported here support the conclusion that GSH is likely used as a nutrient source for *R. parkeri*, which promotes its cell division, actin-based motility, and innate immune evasion. It is also required for growth in Vero cells and is especially important for survival in primary macrophages. The mechanisms in *Rickettsia* for GSH uptake, GSH breakdown to cysteine, and the connection of GSH to cell division remain unclear and warrant

future investigations. Our findings contribute to the paradigm that GSH usage by bacterial pathogens is complex and multifaceted, whereby certain pathogens rely on GSH as a nutrient source or protection from ROS, while others use it as a cue to sense the cytosol and initiate virulence, and some pathogens use GSH for multiple processes. These wide-ranging roles for GSH highlight the critical importance of this metabolite in the pathogenesis and survival of intracellular bacteria.

The machinery encoded by *Rickettsia* to import, transfer, or degrade GSH has not been well described. *Francisella tularensis* acquires GSH and degrades it with the γ-glutamyltranspeptidase GGT and the γ-glutamylcyclotransferase ChaC[21]; however, using BLAST we found that GGT and ChaC homologs are not encoded in *R. parkeri* or related species. *F. tularensis* also synthesizes GSH with the enzymes GshA and GshB, yet we also found no conserved homologs in *R. parkeri* or related SFG *Rickettsia*. GSH transporters are widely conserved among bacteria, including potential homologs of GsiA from the *Escherichia coli* and *Salmonella enterica*[53–55], GshT in Gram-positive *Streptococcus mutans*[56], and the Ctp/OppDF complex in *L. monocytogenes*, which is similar to *E. coli* GsiA[57]. We found that *R. parkeri* encodes an ABC-type transporter (MC1_00730) with 33% identify to *E. coli* GsiA, although its function remains unknown. We and others also observed that *Rickettsia* species encode a conserved GSH S-transferase GstA[10], which is a homolog of a cell division protein FzlA that adopts a GST fold but does not bind GSH in vitro[58,59]. A previous transposon mutagenesis in *R. parkeri* identified a transposon insertion mutant in *gstA* (*Mc1_03795*, insertion site 668,038)[44]. We found that this mutant does not have a cell division defect (Supplemental Fig. 3); however, the transposon insertion is at the far C terminus, so it is unclear if this is a functional knockout. Future studies on GSH transport and breakdown are required to better understand the mechanisms by which GSH and cysteine contributes to *Rickettsia* cell division.

As *Rickettsia* cannot grow in defined or any known cell-free media, it is distinctly challenging to parse apart the roles for individual metabolites in bacterial processes. Detailed bioinformatic studies demonstrated that *Rickettsia* lack the ability to synthesize >50 metabolites[10], which is significantly more than facultative pathogens and even other obligate intracellular bacteria such as *Chlamydia*[60]. Our observation that *R. parkeri* can still grow in GSH-depleted cells in chains, and the observation that NAC can rescue growth in BSO-treated cells, suggests that GSH may not be the lone cysteine source. Previous studies found that in a cell-free media (that does not support growth), cysteine could partially substitute for GSH to promote *Rickettsia* infectivity[61], aligning with our studies. Developing an axenic media would be a valuable tool for the field to enable improved studies into *Rickettsia* metabolism, nutrient uptake, and pathogenesis.

We observe that GSH depletion causes *R. parkeri* to chain. Filamentation is associated with a shared cytosol and is caused by a variety of stressors including DNA damage (the SOS response), mediated by RecA. RecA binds single stranded DNA, eliciting transcriptional changes that can inhibit proper cell division[31]. Interestingly, *L. monocytogenes spx1A* mutants form filaments in liquid media or inside cells[62]. Spx1A is a widely conserved transcriptional regulator that induces expression of peroxide-detoxifying genes. The filamentation of *L. monocytogenes* is likely independent of RecA. Our findings that individual rod-shaped bacteria appear within chains upon GSH depletion are therefore distinct from these previous findings on RecA/SOS and Spx1A. We hypothesize that GSH may promote *R. parkeri* peptidoglycan remodeling via a cysteine requirement of periplasmic proteins such as autolysins, or via a specific role for cysteine in regulating autolysin gene expression, which will require further investigation.

*R. parkeri* evade ubiquitin targeting and antibacterial autophagy via multiple mechanisms, which are required for pathogenesis and causing disease in vivo. *R. parkeri* protein methyltransferases 1 and 2 (PkmT1 and PkmT2) methylate the lysines of surface outer membrane proteins (OMPs), which blocks ubiquitin targeting of these same lysines[44,45]. O-antigen is also required for *R. parkeri* to evade ubiquitin targeting[44], as is the patatin-like phospholipase Pat1[46]. However, the relationship between these mechanisms and our observation that GSH is required to evade polyubiquitylation remains unclear. Future studies on the relationship between GSH and PkmT1/2, OMPs, O-antigen, and Pat1 may reveal the mechanisms by which host nutrient sensing protects against targeting by innate immunity.

We observed that BSO treatment altered *R. parkeri* actin-based motility. Unlike other cytosolic pathogens such as *Listeria monocytogenes*, *Burkholderia thailandensis*, *Mycobacterium marinum*, or *Shigella flexneri* that have one mode of actin-based motility[63], *R. parkeri* undergoes two temporally distinct phases of actin-based motility. The early phase (5–15 min) is mediated by RickA and results in short, curly actin tails, while a later phase (8–72 hpi) mediated by the formin-like protein Sca2 results in long tails[34]. Both RickA and Sca2 contribute to cell-to-cell spread, however Sca2-based motility is the major mediator of cell-to-cell spread[34–37,64]. As we observed actin polymerization at the poles of chained bacteria, our findings support the hypothesis that GSH is required for cell division, which then indirectly impacts actin-based motility. An alternative hypothesis could be that GSH is directly required for Sca2-mediated actin-based motility, as we observed that *R. parkeri* lacked actin association at their surfaces later in the infection. Further experiments will be required to tease apart the role for nutrient acquisition in promoting actin-based motility.

The growing consensus that many phylogenetically distinct intracellular pathogens rely on eukaryotic GSH for intracellular growth and survival may suggest that the reducing cytosol is a host-associated molecular pattern (HAMP). Whereas pathogen-associated molecular patterns (PAMPs) are conserved molecular features of pathogens that are sensed by host cells as a signature of infection[65], GSH may be a HAMP that is detected by pathogens to promote their virulence. Detection of PAMPs by host pattern recognition receptors (PRRs) leads to a host defense response including upregulation of antimicrobial genes and programmed cell death. The receptors, ligands, and downstream signaling responses of PAMPs and PRRs are well characterized[66]; however, HAMPs are less well understood. Identifying and revealing the molecular mechanisms of how pathogens sense and uptake eukaryotic metabolites will increase our understanding of bacterial pathogenesis and potentially lead to new strategies to therapeutically target host metabolism as an antimicrobial strategy.

## Methods

### Ethics statement
This study complies with all relevant ethical regulations, including the approved biological use authorization (Burke lab BUA-R451). Animal research using mice was conducted under a protocol approved by the UC Irvine Institutional Animal Care and Use Committee (IACUC) in compliance with the Animal Welfare Act and other federal statutes relating to animals and experiments using animals (Burke lab animal use protocol AUP-22-005 and AUP-25-004). The UC Irvine IACUC is fully accredited by the Association for the Assessment and Accreditation of Laboratory Animal Care International and adheres to the principles of the Guide for the Care and use of Laboratory Animals. Mice were housed in standard 7 am–7 pm light/dark cycles under specific pathogen free (SPF) conditions with ambient temperature and humidity.

### Preparation of *R. parkeri*
*R. parkeri* strain Portsmouth was originally obtained from Dr. Christopher Paddock (Centers for Disease Control and Prevention). Bacteria were amplified by infecting confluent T175 flasks of female African green monkey kidney epithelial Vero cells. These were originally obtained from the UC Berkeley Cell Culture Facility were authenticated

by mass spectrometry experiments. Vero and HMEC cells were tested routinely for mycoplasma contamination (ABCAM Ab289834). Vero cells were grown in DMEM (Gibco 11965-092) with glucose (4.5 g/L) supplemented with 2% fetal bovine serum (FBS, Corning 35-010-CV). To prepare bacteria for infection, T175 flasks of confluent Vero cells were infected with $5 \times 10^6$ R. parkeri per flask and at 5 dpi when cells were heavily infected, cells were scraped, collected, and centrifuged at 12,000 x G for 20 min at 4 °C. Pelleted cells were then resuspended in K-36 buffer (0.05 M KH$_2$PO$_4$, 0.05 M K$_2$HPO$_4$, 100 mM KCl, 15 mM NaCl, pH 7) and dounced (60 strokes) on ice. The solution was then centrifuged at 200 x G for 2 min at 4 °C to pellet host cell debris. Supernatant containing R. parkeri was overlaid on a 30% Ficoll (Cytiva, 17144002) solution in K-36 buffer. Gradients were centrifuged at 18,000 rpm in an SW-28 ultracentrifuge swinging bucket rotor (Beckman/Coulter) for 20 min at 4 °C to separate host cells debris. Bacterial pellets were resuspended in brain heart infusion (BHI) media (BD, 237500) and stored at −80 °C. Bacterial titers were determined via plaque assays by serially diluting the bacteria in 12-well plates containing confluent Vero cells. Plates were then spun for 5 min at 300 x G in an Eppendorf 5810 R centrifuge. Titers were determined using the plaque assay described below.

## Deriving bone marrow macrophages

To obtain bone marrow, mice were euthanized, and femurs, tibias, and fibulas were excised. Combined data from multiple experiments include BMDMs from both male and female mice. Connective tissue was removed, and the bones were sterilized with 70% ethanol. Bones were washed with BMDM media (20% Corning FBS, 1% sodium pyruvate, 0.1% β-mercaptoethanol, 10% conditioned supernatant from 3T3 fibroblasts, in Gibco DMEM containing glucose and 100 U/ml penicillin and 100 ug/ml streptomycin) and ground using a mortar and pestle. Bone homogenate was passed through a 70 μm nylon Corning Falcon cell strainer (Falcon 352350) to remove particulates. Filtrates were centrifuged in an Eppendorf 5810 R at 1200 RPM (290 x G) for 8 min, supernatant was aspirated, and the remaining pellet was resuspended in BMDM media. Cells were then plated in non-TC-treated 15 cm petri dishes (at a ratio of 10 dishes per 2 femurs/tibias) in 30 ml BMDM media and incubated at 37 °C. An additional 30 ml media without antibiotics was added 3 d later. At 7 d the media was aspirated, and cells were incubated at 4°C with 15 ml cold PBS (Gibco, 10010-023) for 10 min. BMDMs were then scraped from the plate, collected in a 50 ml conical tube, and centrifuged at 1200 RPM (290 x G) for 5 min. PBS was then aspirated, and cells were resuspended in BMDM media with 30% FBS and 10% DMSO at $1.2 \times 10^7$ cells/ml. 1 ml aliquots were stored in liquid nitrogen.

## In vitro experiments

To plate cells for infection, aliquots of BMDMs were thawed on ice, diluted into 9 ml of DMEM, centrifuged in an Eppendorf 5810 R at 1,200 RPM (290 x G) for 5 min, and the pellet was resuspended in 10 ml BMDM media without antibiotics. The number of cells was counted using Trypan blue (Sigma, T8154) and a hemocytometer (Bright-Line), and $5 \times 10^5$ cells were plated into 24-well plates. Approximately 16 h later, 30% prep R. parkeri were thawed on ice and diluted into fresh BMDM media to the desired concentration. Media was then aspirated from the BMDMs, replaced with 500 μl media containing R. parkeri, and plates were spun at 300 G for 5 min in an Eppendorf 5810 R. Infected cells were then incubated in a humidified CEDCO 1600 incubator set to 33 °C and 5% CO$_2$ for the duration of the experiment. L-BSO was obtained from Sigma (B2515) and was added to cells after plating the day before infections. BSO was removed during infections and replaced 30 mpi.

Titers were determined via plaque assays by serially diluting the bacteria in 12-well plates containing confluent Vero cells. Plates were then spun for 5 min at 300 x G in an Eppendorf 5810 R centrifuge. At

24 hpi, the media from each well was aspirated and the wells were overlaid with 2 mL/well DMEM with 5% FBS and 1.2% Avicel® PH-101 (Sigma,11363). At 7 dpi, the wells were fixed with 2 mL/well of 7% Formaldehyde (VWR,10790-708) for 30 min and with 0.5 mL/well of 1x crystal violet (VWR, 470337-534) for 15 min. Crystal violet was then washed off with tap water and plaques were counted.

For the resazurin viability assay, $10^5$ Vero cells were plated per well in clear bottom 96-well plates. Resazurin sodium salt (Fisher, AAB2118703) was added to each well at 44 μM. Cells were cultured for 2 h and then read using a ClarioStar-Plus plate reader at absorbances of 570 and 595 nm. The final absorbance was calculated by subtracting absorbance of 595 nm from that of 570 nm.

For experiments involving L-NIL, $5 \times 10^5$ WT, Atg5$^{fl/fl}$ and Atg5$^{-/-}$ BMDM were plated per well in 24 well plates. 2 mM BSO was added overnight. The following day, media was aspirated, cells were infected with R. parkeri at a MOI of 1, and BSO was replaced at 1 hpi. 1 mM L-NIL (MedChemExpress, HY-12116) was added to each well 1 h postinfection. BMDMs were then lysed at different timepoints and plated on Veros for quantification via PFU assay.

For the RAW Blue NF-κB assay, $10^5$ RAW Blue cells (Invivogen, raw-sp) were plated per well in clear bottom 96-well plates. The following day, cells were infected with R. parkeri at a MOI of 1. Cells were then cultured for 48 h and supernatants were collected. NF-κB activation was measured using QUANTI-Blue™ solution (Invivogen, rep-qbs). 180 μL of QUANTI-Blue™ solution was added per well in a clear bottom 96 well plate, and then 20 μL of the supernatant was added per well. The plate was then incubated for 2 h at 33 °C and then read using ClarioStar-Plus plate reader at 620 nM.

For measuring ROS production, $5 \times 10^4$ Veros cells were plated in black/clear bottom 96 well Thermo (165305) plates overnight. The cellular ROS detection kit (Abcam, AB113851) was used according to manufacturer's instructions. Briefly, the following day, cells were treated with DCFH-DA at 20 μM for 45 min and then washed, and then treated with the indicated conditions for 4 h. The positive control was 100 μM TBHP solution. Cells were then read using the ClarioStar-Plus plate reader at excitation/emission at 485/535 nm.

For experiments with metabolites or autophagy inhibitors, $5 \times 10^5$ WT BMDM were plated per well in 24 well plates. 2 mM BSO was added overnight (final concentration). The following day, media was aspirated, cells were infected with R. parkeri at a MOI of 1, and BSO was replaced at 1 hpi. 2 mM BSO, 2 mM L-glutathione (Thermo Scientific, J6216606), 5 mM N-acetylcysteine (Tocris, 7874), 50 μM L-cysteinyl-glycine (Sigma, C0166), 5 mM GSHee (Cayman, 14953), and 20 μM H$_2$O$_2$ (Fisher, BP2633500) was added to each well 1-hpi (final concentrations). For autophagy inhibitor experiments, 3-methyladenine (3MA, Selleck, S2767) was added to each well 1-hpi to a final concentration of 5 mM. BMDMs were then lysed at different timepoints and plated on Veros for quantification via PFU assay.

For LDH assays, 60 μl of supernatant from wells containing BMDMs was collected into 96-well plates and LDH assay buffer was added according to manufacturer's instructions (Promega CytoTox 96 Non-Radioactive Cytotoxicity Assay, G1780). Supernatant from uninfected cells was used as a negative control and from cells lysed with 1% triton X-100 (final concentration) was a positive control. Reactions were incubated at room temperature for 30 min prior to reading at 490 nm using a ClarioStar-Plus plate reader. Values for uninfected cells were subtracted from the experimental values, divided by the difference of triton-lysed and uninfected cells, and multiplied by 100 to obtain percent lysis. Each experiment was performed and averaged between technical duplicates and biological triplicates.

For the IFN-I bioassay, $5 \times 10^4$ 3T3 cells containing an interferon-sensitive response element (ISRE) fused to luciferase[51] were plated per well into 96-well white-bottom plates (Greneir 655083) in DMEM containing 10% FBS, 100 U/ml penicillin and 100 μg/ml streptomycin. Media was replaced 24 h later and confluent cells were treated with 2 μl

of supernatant harvested from BMDM experiments. Media was removed 4 h later and cells were lysed with 40 μl TNT lysis buffer (20 mM Tris, pH 8, 200 mM NaCl, 1% triton-100). Lysates were then injected with 40 μl firefly luciferin substrate and luminescence was measured using the luminometer function of the ClarioStar-Plus plate reader.

## Microscopy

For immunofluorescence microscopy, either 1 or $2.5 \times 10^5$ Vero, HMEC-1 (originally obtained from Dr. Matt Welch, UC Berkeley), or BMDMs were plated overnight in 24-well plates on sterile 12 mm coverslips (Thermo Fisher Scientific, 12-545-80). Infections were performed as described above. At the indicated times post-infection, coverslips were washed once with PBS and fixed in 4% paraformaldehyde (Ted Pella Inc., 18505, diluted in 1 x PBS) for 10 min at room temperature. Coverslips were then washed 3 times in PBS. Coverslips were washed once in blocking buffer (1 x PBS with 2% BSA) and permeabilized with 0.5% triton X-100 for 10 min. Coverslips were incubated with primary or secondary antibodies diluted in 2% BSA in PBS for 30 min at room temperature. *R. parkeri* was detected using anti-*Rickettsia* 14-13 (mouse) or I7205 (rabbit) (originally from Dr. Ted Hackstadt, NIH/NIAID Rocky Mountain Laboratories), or antibody specific for *Rickettsia japonica* OMP (Invitrogen, PA5-117668). All coverslips were stained with 14-13, except for Fig. 1f (*R. japonica* OMP antibody from Invitrogen), and Fig. 4a (I7205). Polyubiquitin was stained the FK1 antibody (Novus, NBP3-11039 lot MU188322) diluted 1:250 in PBS with BSA. Nuclei were stained with DAPI (BD Biosciences 564907) diluted 1:2,000, and actin was stained with Alexa-568 phalloidin (Life Technologies, A12380) diluted 1:500. The I7205, 14-13, FK1 antibodies were previously validated in similar studies using no-infection controls or controls with no ubiquitylation[45]. Secondary antibodies were AlexaFluor-488 goat anti-mouse (Invitrogen A11001, lot 2513496), AlexaFluor-647 goat anti-rabbit (Invitrogen A32733, lot XG349344), AlexaFluor-488 goat anti-rabbit (Invitrogen A11008, lot 2420731) all diluted 1:500. Coverslips were mounted in ProLong™ Gold Antifade Mountant (Invitrogen, P36934). Samples were imaged with either the Keyence BZ-X810 Inverted Microscope with 20x or a 60x oil objective or the Airyscan 2/ GaAsP function of the Zeiss LSM 900 Confocal Multiplex with Plan-Apochromat 63x/1.4 Oil DIC M27 (FWD = 0.19 mm) objective or the Plan-Apochromat 20x/0.8 M27 (FWD = 0.55 mm) objective at the UC Irvine Microscope Imaging Core. Images were processed using (ImageJ) FIJI[49] version 2.16.0/1.54p and brightness and contrast adjustments were applied to entire images. Images were assembled using Adobe Illustrator. Representative images are a single optical section. For quantification and cell length measurements, images were blinded. Images were analyzed using FIJI, descriptions of the number of bacteria counted are described in figure legends.

## Statistical analysis

Statistical parameters and significance are reported in the figure legends. For comparing two sets of data, a two-tailed Student's *T*-test was performed. For comparing multiple data sets, a one-way ANOVA with multiple comparisons with Tukey post-hoc test was used for normal distributions, and a Mann-Whitney *U* test was used for non-normal distributions. Sample sizes were chosen based on existing literature in the field for similar experiments[39,43]. No animals were used in this study and thus organisms were not allocated into experimental groups. Data are determined to be statistically significant when $p < 0.05$. Asterisks denote statistical significance as: *$p < 0.05$; **, $p < 0.01$; ***$p < 0.001$; ****$p < 0.0001$, compared to indicated controls. Error bars indicate standard deviation (SD). All other graphical representations are described in the Figure legends. Statistical analyses were performed using GraphPad PRISM version 10.4.1. All *P*-values are reported in Supplemental Table 1.

## Reporting summary

Further information on research design is available in the Nature Portfolio Reporting Summary linked to this article.

## Data availability

*R. parkeri* strains were authenticated by whole genome sequencing; the sequencing data are available in the NCBI Sequence Read Archive (SRA) under accession number SRX4401164. Source data are provided in the Dryad repository https://doi.org/10.5061/dryad.905qfttvz. Source data are provided with this paper.

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

## Acknowledgements

WT *R. parkeri* was a generous gift from Dr. Matthew Welch (University of California, Berkeley). The AausFP1-expressing *R. parkeri* was a generous gift from Dr. Erin Goley (Johns Hopkins University). *Atg5*[-/-] and *Atg5*[fl/fl] cells were a generous gift from Dr. Christina Stallings (Washington University). M.D. was supported in part by ACS Seed Grant 129801-IRG-16-187-13-IRG from the American Cancer Society. M.D. was supported in part by the 5T32AI177234 Immunology Research Training Grant.

## Author contributions

H.S., A.P.L, M.D., and T.P.B. performed and analyzed experiments. T.P.B. wrote the original draft of the manuscript. Critical reading and further edits were also provided by H.S., M.D., and A.P.L. Supervision was provided by T.P.B.

## Competing interests

The authors declare no competing interests.
