## [Transparent Peer Review file · Nature Communications]

Host glutathione is required for *Rickettsia parkeri* cell division and intracellular survival.

Corresponding Author: Dr Thomas Burke

Version 1:

Reviewer comments:

Reviewer #1

(Remarks to the Author)

Glutathione (GSH) was predicted to be imported by *Rickettsia* since they do not have the ability to synthesize GSH but do have a glutathione S-transferase to transfer GSH to substrates. In this study, Sun et al. investigate the role of GSH in *Rickettsia parkeri* infection by inhibiting host GSH synthesis with the small molecule BSO. Specifically, they have explored the effect of GSH-depletion on bacterial shape, actin localization, bacterial and host cell lysis, and innate immune response in a variety of cell types and produced really beautiful microscopy images of interesting bacterial cell shape morphologies and actin localization. They conclude that GSH is critical for *Rickettsia* septation, actin-based motility, evading autophagy, and survival in immune cells.

While the high quality data presented by the authors demonstrate multiple phenotypes resulting from GSH-depletion, I think they could go a step further towards identifying a mechanistic explanation for some of the phenotypes. It is not realistic to determine the mechanism for each phenotype, but the authors should attempt to narrow it down at least by assessing if GSH is used as nutrient (versus signaling molecule) and to consider more of the possible scenarios in the Discussion to make their discoveries more accessible to a broad audience. Specific comments below.

Major Comments

1. While the authors conclusively demonstrate that GSH is important for *Rickettsia*, it's not clear if it is being used as a nutrient and/or a signaling molecule. It's role in signal transduction would be complex to disentangle, but it should be feasible to determine if it is being used as a cysteine source, which the following evidence would support:

- There is obviously a more pronounced growth defect in BMDMs than in Vero cells, but it looks like there are fewer bacteria in the BSO-treated Vero cells (Fig 1B), although this was not quantified. The number of bacteria in Fig 1B could easily be enumerated by microscopy by counting bacteria per field of view or bacteria per cell. This would enable a more direct comparison between Vero cells and BMDMs, which the authors discuss several times.
- Although Atg5-deletion rescues about 1-log of bacterial death in the presence of BSO, the bacteria still do not replicate over 72 hours in this condition.
- The citations used to support the authors' conclusion that GSH is not the sole cysteine source (lines 195-197) actually conclude that *Rickettsia* can import S-adenosylmethionine to use in the synthesis of polyamines, not as a source of cysteine (39). I did not find mention of cysteine in citation 10. Perhaps the authors missed a citation?

Together, these data are consistent with the hypothesis that GSH is a significant (but not sole) cysteine/sulfur source for *Rickettsia* and they are not replicating as well in BSO-treated cells without this important nutrient. To address this, the authors could try rescuing BSO-treated cells with excess GSH, Cys-Gly dipeptide, cysteine, NAC, or any other sulfur source. If additional sulfur does not rescue the BSO-dependent phenotypes, that would go a long way to strengthen the argument that these "cytosolic bacteria are sensing redox to cue for specific aspects of virulence" as mentioned throughout the text, and not simply using it as a nutrient.

2. The Discussion needs more description of *Rickettsia* actin recruitment to make it accessible to a wide audience. What are the differences between straight and curly tails produced by Sca2 and RickA, respectively? How are these things regulated? How might GSH be important for these processes?

Minor Comments

- line 102: it doesn't really make sense to say "frequency of absent actin"
- Fig 2A: not a lot of actin tails are visible in this image

- Fig 2E and F: These data do not really add to the conclusions of the paper and should be removed. The alignment is not necessary since this protein was not investigated in this study and the microscopy is not as detailed or informative as the other images.
- Fig 3B: it's very hard to see the bacteria here. I realize it is hard since there are not many surviving bacteria at this time point, but it may help to include multiple panels of examples
- Fig 3D: is there a significant growth defect in Casp1/11-/- +BSO similar to what is seen in WT+BSO? If so, the fewer bacteria may explain less IFN. This should be easy to quantify.
- line 209: missing the word 'factor' or 'regulator' after transcription
- the text could be more descriptive throughout. For example, lines 145-150 describe that lysis of *R. parkeri* leads to pyroptosis and then LDH is measured. It is assumed that the reader understands that pyroptosis leads to LDH release, which is what is measured as a proxy for pyroptosis. This should be clearly stated. Similarly for RLUs as a measurement of IFN release.

Reviewer #2

(Remarks to the Author)

In their manuscript, Sun et al describe the phenotypic outcomes of depletion of glutathione on intracellular replication of the obligate intracellular human pathogen, *Rickettsia parkeri*. *Rickettsia* species have undergone reductive evolution, leaving them with streamlined genomes and a reliance on a host cell for numerous essential metabolites. Glutathione has been implicated in virulence of a number of pathogens, motivating the authors to examine the role of glutathione in growth and pathogenesis of *R. parkeri*. Treatment of host cells with buthione sulfoximine (BSO) to deplete glutathione during *R. parkeri* infection caused a number of defects, including reduced plaque size/number, cell division defects, reduced actin-based motility, and an inability to avoid autophagy. The authors propose that glutathione is required to support *R. parkeri* intracellular growth, perhaps through its role in redox homeostasis.

Overall this manuscript describes novel connections between a host metabolite and *R. parkeri* biology, although mechanisms linking glutathione to the phenotypes observed are not yet clear. Except for a few places noted below, the conclusions are largely supported by the data and the findings should be of interest to those interested in *Rickettsia* biology and host-pathogen interactions. The authors may want to consider the following suggestions to strengthen their manuscript:

1. The perturbation used to impact glutathione throughout the study is treatment with BSO. Does BSO impact host cell viability or replication?
2. The authors report a "septation" defect with BSO treatment, which they define as cell chaining. The description and quantitation of this phenotype could be improved as follows:
 - a. The authors suggest that glutathione depletion causes bacterial cell chaining, which would be a cell separation defect, not a septation defect. Septation refers to formation of a septum, which the authors suggest does occur in BSO-treated *R. parkeri*.
 - b. It is not clear from the images provided whether the cell division defect is actually in cell separation or in formation of a septum (or both). Some cells do look chained, but others look more purely filamentous (without septa), in Figure 1C for example. Use of a cytoplasmic marker might be a more direct way to assess if the cytoplasm is compartmentalized in the filamentous cells. Alternatively, or in addition, electron microscopy would reveal the extent to which cells have (or do not have) a septum. Understanding which stage(s) of cell division is impacted would help inform hypotheses for how glutathione affects cell division.
 - c. From the images in Figure 1B, the cell division defect appears to be at its worst at ~48 hpi, with cells becoming shorter or more heterogeneous in length at 72 and 96 hpi. Is this the case? Can the authors quantify cell length as a function of time post infection across the population? It could be that cells are acquiring suppressors or otherwise overcoming the effects of glutathione depletion over time. Related to this, the authors report approximate cell lengths in the text - these analyses would be strengthened by reporting cell length for populations of treated and untreated cells.
3. Lines 112-114. The effects of glutathione depletion on actin-based motility seem indirect, given the extended treatment time required to observe an effect. Do the authors think this is an indirect effect of the cell division defects they observe?
4. Lines 123-128. The authors indicate that a putative glutathione S-transferase is encoded in *Rickettsia* species. The protein/gene mentioned actually appears to be a homolog of a cell division protein called FzIA that has been well-described in *Caulobacter crescentus* and, to a lesser extent, *Agrobacterium tumefaciens* (Goley et al 2010 Mol Cell, Lariviere et al 2018 Mol Micro, Lariviere et al 2019 Curr Biol, Mahone et al 2024 JCB). The *Caulobacter* FzIA homolog adopts a GST fold, but apparently does not bind glutathione in vitro - however, it seems possible that the *R. parkeri* protein might? If so, this could be a potential link between glutathione and cell division that should be mentioned, if not tested. Moreover, the transposon insertion in the gene encoding the *R. parkeri* FzIA homolog would disrupt the extreme C-terminus of the protein, so it may not be a complete loss of function. It is also possible that there are suppressing mutations elsewhere in the genome.
5. Lines 24 and 70-71: The authors conclude that their data support a paradigm of pathogens sensing host redox state as a cue for virulence, but this is not directly addressed experimentally. Given the other potential effects of glutathione on metabolism or protein function, etc, the authors might temper their conclusions to this effect. In addition, the effects reported here seem less effects on virulence, per se, and more on bacterial physiology (as the authors note in the discussion, lines 193-195).
6. Line 31, "similar genome to *R. parkeri*" should be "similar genome to *R. rickettsii*"

Reviewer #3

(Remarks to the Author)
Significance

The manuscript entitled "Host glutathione is required for *Rickettsia parkeri* to properly septate, avoid ubiquitylation, and survive in macrophages" tries to establish a link between host cell glutathione and how it modulates *Rickettsia parkeri* intracellular growth and survival. Overall, the manuscript is well written and fairly easy to follow. The experimental approaches are based on published data that is both from the author's group and other investigators. However, one of the main weaknesses of this work is that it provides a series of interesting observations, but does not address the mechanism(s) as to how *Rickettsia parkeri* utilizes host glutathione for the intracellular growth and dissemination.

Specifically:

The authors need to show how (or if) *R. parkeri* imports glutathione from the host. What is the mechanism of glutathione utilization by rickettsial metabolic pathways to support obligate intracellular life?

The authors just presented phenotypic data of rickettsial survival, septation, actin-based motility, ubiquitylation, autophagy in depleting glutathione using buthionine sulfoximine (BSO).

No experimental evidence of glutathione levels in the host cells (uninfected, infected or with BSO treatment) was presented. This is surprising as 1) it is well known that BSO treatment modulates such levels and 2) would have helped to strengthen the significance of the authors findings. The authors should provide data to why the authors chose the time and concentration of BSO utilized in the paper.

What are the endogenous glutathione levels of the utilized cells and are they different among the cell types (epithelial, endothelial and macrophages)? Could such potential differences in endogenous glutathione levels help explaining the phenotypical differences observed in these cell types?

The authors attempt to show a connection between glutathione metabolism and *R. parkeri* survival requires more rigorous examination. The authors should provide additional insights into how BSO treatment modulates other host cytokine responses (currently only focusing on IFN-I response) and how *R. parkeri* infection (+/- BSO treatment) modulates ROS production.

Also, it is unknown whether the presented results (Figure 4) are related to Atg5-dependent autophagy or only ATG5 itself. Especially because ATG5-mediated host event independent of autophagy has been published previously.

The authors should further address the mechanism of how rickettsial molecules/effectors become engaged in utilizing host glutathione to support intracellular growth and dissemination.

Version 2:

Reviewer comments:

Reviewer #1

(Remarks to the Author)

The authors have made significant progress and satisfactorily addressed all comments from the previous submission. The inclusion of new data strongly supports the conclusion that GSH serves as the primary source of sulfur/cysteine for cytoplasmic *Rickettsia*. However, the revised order of data presentation is somewhat confusing, and the interpretation of some data could be clearer. I recommend reorganizing the manuscript to integrate the new data into a more cohesive narrative. For example, Figures 3A-D could be combined with Figure 1A, as they all pertain to cell-cell spread. Similarly, replication and morphology defects could be consolidated into a single (or 2) figure(s) for clarity. Of course, the authors are free to structure the manuscript as they see fit, but the current arrangement reads more as a series of phenotypes rather than emphasizing the mechanistic insights. This approach obscures the depth of their work, which deserves to be highlighted.

In addition, using more precise language would strengthen the manuscript. For instance, the statement on line 244, "this is the first report linking the uptake of eukaryotic GSH to bacterial cell division," implies that GSH is specifically linked to cell division. A more accurate phrasing would emphasize that GSH is required for bacterial replication. Improving language clarity will help to better convey the central conclusion: GSH serves as an essential nutrient for *Rickettsia*.

Specific Comments

Fig 1C does not appear to me to be a quantification of B. The BSO-treated line in C does not increase at all over 96 hours, suggesting ~ 2 bacteria per cell. Conversely, the images in C show a clear increase in the number of bacteria over time and far more than 2 bacteria in each cell shown.

Importantly, 1C and 1D were not mentioned in the text, but these data clearly show a replication defect, which could explain

the plaque defect. But the text ignores those results and instead investigates actin tails as an explanation for the plaque defect.

Fig 2: the addition of Fig 2D is really nice

Line 124 should call out Fig 3B

Line 127 should call out Fig 3D

Fig 6: the images in B are not very informative as they are too small and zoomed out to see bacterial morphology. The quantification is sufficient so I would suggest eliminating the images to allow panels C and D to be bigger and more comprehensible.

Reviewer #2

(Remarks to the Author)

In their revised article, Sun and colleagues have provided additional data supporting a role of host glutathione (GSH) in promoting growth and division of *Rickettsia parkeri* as a source of cysteine. They demonstrate that treating infected cells with an inhibitor of host glutathione synthesis causes defects in growth, cell division, and autophagy evasion. Supplying cells with alternative cysteine sources rescues these defects.

Overall the authors have addressed my major concerns from their initial submission. The inclusion of data to support a role of glutathione as a nutrient source helps to provide mechanistic insight into why depletion of GSH is deleterious. I have minor suggestions for improvement of the clarity of the manuscript.

1. Line 17 and 70. Instead of "chaining and survival were restored", it should be "cell division and survival were restored".
2. Lines 71-73. "...critical role for host GSH in *Rickettsia* cell division pathogenesis and contribute..." should be edited for clarity. Do you mean cell division and pathogenesis?
3. Line 87. Change "...if BSO caused toxicity in our culture system..." to "if BSO caused toxicity to host cells in our system..." for clarity.
4. Line 96. The reference to Fig 1B,C seems like it should be to Fig. 2B,C since that illustrates cell chaining. In general, it would be preferred to arrange figures and figure panels in the order to which they are referred in the text.
5. Line 99. Add reference to Fig 1D, 2C.
6. Line 122. "nucleated actin" would be better as "filamentous actin" since phalloidin specifically reports on F-actin.
7. Line 127. The image of HMEC cells shows *R. parkeri* with clouds... did any bacteria have tails in HMEC cells?
8. Line 189. Did the authors test if addition of GSH itself restored survival? That data was not shown, or I could not find it. If it does not, do the authors have an explanation for that? Can host cells not metabolize GSH to GSHee for import? Similarly, if cystine is the form of cysteine normally imported to *Rickettsia* can the authors explain why added cystine does not restore division and growth? That would seem to call into question the use of GSH as a cysteine source.
9. For figure 6, consider re-ordering the panels so they are presented in the order they are described in the text.
10. Line 308. "Sca2 mediated" should have a hyphen.
11. Fig 1 and legend, add what is labeled/visualized in panel B to the legend and, if possible, the image. (What is green, what is red?)
12. Figs 2, 3, 4, 6, S2. Again it would be helpful for clarity to indicate on the images what each panel is showing (actin, bacteria, etc). The legends for Fig 3 and 4 also need a description of what is labeled. Also, the order of panels in Fig 3A is different than in B and D (i.e. green is shown first in A, 2nd in B and D). Consider making them consistent throughout the figure.

Reviewer #4

(Remarks to the Author)

In the revised manuscript, the authors presented additional data from new experiments supporting the role of GSH as a cysteine nutrient source. All concerns raised by this reviewer are addressed. There are two minor typos that need to be fixed.

Line 306: Should read as " cell division, which then...."

Line 309: Should read "surfaces later in the infection."

Response to Reviewers

Sun et al., 2024

Overall:

We thank the reviewers for their valuable critiques and input. We have sought to address all the reviewer concerns, and the revised manuscript has 2 new main text figures and 4 additional supplemental figures. The additional experiments have strengthened the paper and added important mechanistic details into understanding how GSH contributes to *Rickettsia* intracellular survival and cell division. Responses to each comment are below.

Reviewer #1 (Remarks to the Author):

Glutathione (GSH) was predicted to be imported by *Rickettsia* since they do not have the ability to synthesize GSH but do have a glutathione S-transferase to transfer GSH to substrates. In this study, Sun et al. investigate the role of GSH in *Rickettsia parkeri* infection by inhibiting host GSH synthesis with the small molecule BSO. Specifically, they have explored the effect of GSH-depletion on bacterial shape, actin localization, bacterial and host cell lysis, and innate immune response in a variety of cell types and produced really beautiful microscopy images of interesting bacterial cell shape morphologies and actin localization. They conclude that GSH is critical for *Rickettsia* septation, actin-based motility, evading autophagy, and survival in immune cells. While the high quality data presented by the authors demonstrate multiple phenotypes resulting from GSH-depletion, I think they could go a step further towards identifying a mechanistic explanation for some of the phenotypes. It is not realistic to determine the mechanism for each phenotype, but the authors should attempt to narrow it down at least by assessing if GSH is used as nutrient (versus signaling molecule) and to consider more of the possible scenarios in the Discussion to make their discoveries more accessible to a broad audience. Specific comments below.

Major Comments

1. While the authors conclusively demonstrate that GSH is important for *Rickettsia*, it's not clear if it is being used as a nutrient and/or a signaling molecule. It's role in signal transduction would be complex to disentangle, but it should be feasible to determine if it is being used as a cysteine source, which the following evidence would support:

i) There is obviously a more pronounced growth defect in BMDMs than in Vero cells, but it looks like there are fewer bacteria in the BSO-treated Vero cells (Fig 1B), although this was not quantified. The number of bacteria in Fig 1B could easily be enumerated by microscopy by counting bacteria per field of view or bacteria per cell. This would enable a more direct comparison between Vero cells and BMDMs, which the authors discuss several times.

We thank the reviewer for this idea. We have now quantified the number of bacteria in Figure 1B and inserted a new panel with the quantification (1C). This reveals that glutathione depletion hinders bacterial abundance over time in Vero cells, which is ~15-fold difference by number of bacteria/cell. This data aligns with the PFU data in Figure 4A showing that there is a 24-fold and 80-fold restriction by PFUs in Vero cells at 48 and 72 hpi, respectively. In comparison, the difference in BMDMs is 1,524-fold at 48 h. We believe that this new data enhances the manuscript and aligns with the conclusion that BMDMs exhibit a more dramatic restriction of *R. parkeri* upon GSH depletion.

ii) Although Atg5-deletion rescues about 1-log of bacterial death in the presence of BSO, the bacteria still do not replicate over 72 hours in this condition.

We thank the reviewer for this observation and agree that *Atg5* deletion does not fully account for the BSO-mediated restriction, which is a rescue of 44-fold at 72 hpi. We have added the following to address the reviewer's concern: "*Atg5* deletion did not fully restore *R. parkeri* growth in GSH-depleted cells, and therefore we conclude that autophagy-mediated restriction plays an important but not complete role in killing the bacteria."

iii) The citations used to support the authors' conclusion that GSH is not the sole cysteine source (lines 195-197) actually conclude that *Rickettsia* can import S-adenosylmethionine to use in the synthesis of polyamines, not as a source of cysteine (39). I did not find mention of cysteine in citation 10. Perhaps the authors missed a citation?

We thank the reviewer for pointing out this oversight, the reviewer is correct that the citation specifies polyamines and not cysteine and we have removed this sentence. We agree that GSH could potentially be used as a cysteine source, which we now discuss throughout the manuscript and the additional data (below) suggests that the bacteria are likely using GSH as a cysteine source.

Together, these data are consistent with the hypothesis that GSH is a significant (but not sole) cysteine/sulfur source for *Rickettsia* and they are not replicating as well in BSO-treated cells without this important nutrient. To address this, the authors could try rescuing BSO-treated cells with excess GSH, Cys-Gly dipeptide, cysteine, NAC, or any other sulfur source. If additional sulfur does not rescue the BSO-dependent phenotypes, that would go a long way to strengthen the argument that these "cytosolic bacteria are sensing redox to cue for specific aspects of virulence" as mentioned throughout the text, and not simply using it as a nutrient.

We thank the reviewer for this suggestion. We performed this experiment and made a number of unexpected observations, including that NAC significantly decreases *R. parkeri* chaining and significantly increases survival in GSH-depleted cells. GSH itself is not imported via host cells but we found that the imported GSH derivative GSHee (GSH ethyl ester) also rescued *R. parkeri* chaining and survival. The Cys-Gly dipeptide did not rescue chaining or survival. Upon adding cysteine, we observed that the amino acids were not soluble and caused the formation of crystal-appearing structures on the plate that were toxic to the host cells, and therefore investigated cystine, which is the known imported version of this amino acid. Cystine was tolerated by the cells however we did not observe any rescue of chaining or survival. In summary, the observation that NAC rescued the chaining phenotypes suggests that GSH is likely used as a nutrient source (Fig. 6).

To expand on this finding, we further investigated whether the GSH or NAC are being used as nutrient sources versus anti-oxidants. We measured bacterial survival and chaining in the presence of hydrogen peroxide, which we found increases intracellular ROS. Interestingly, we did not observe bacterial chaining or bacterial restriction upon increased ROS (Fig. 6). We also measured ROS production elicited by infection and found that it was decreased, and we report that BSO only increases ROS a small amount. These data suggests that requirement for GSH in promoting bacterial cell division and survival is primarily as a nutrient source and likely not mediated through ROS.

2. The Discussion needs more description of *Rickettsia* actin recruitment to make it accessible to a wide audience. What are the differences between straight and curly tails produced by Sca2 and RickA, respectively? How are these things regulated? How might GSH be important for these processes?

We thank the reviewer for these ideas and we have now included a detailed paragraph about Sca2 and RickA in the Discussion, including their regulation and potential role for GSH in actin-based

motility.

Minor Comments

- line 102: it doesn't really make sense to say "frequency of absent actin"

We have changed this text.

- Fig 2A: not a lot of actin tails are visible in this image

We have now included a super-resolution image where the tails are more apparent (Fig. 3A).

- Fig 2E and F: These data do not really add to the conclusions of the paper and should be removed. The alignment is not necessary since this protein was not investigated in this study and the microscopy is not as detailed or informative as the other images.

We agree, and the additional data showing that NAC can rescue *R. parkeri* chaining and survival in BSO-treated cells suggests that GSH is likely not being used as a post-translational modification that impacts intracellular growth. Thus GstA is unlikely to be playing an important role in these processes. Reviewer 2 expressed interest in these factors and therefore they may be of a wider importance to the field, and so we have kept the alignment and performed additional imaging that is higher quality, but we have de-emphasized this finding and moved it to the supplemental.

- Fig 3B: it's very hard to see the bacteria here. I realize it is hard since there are not many surviving bacteria at this time point, but it may help to include multiple panels of examples.

We have zoomed in on this image to better show the bacteria. We now include additional panels showing the infected BMDMs treated with BSO (Fig. 4B).

- Fig 3D: is there a significant growth defect in *Casp1/11*^{-/-} +BSO similar to what is seen in WT+BSO? If so, the fewer bacteria may explain less IFN. This should be easy to quantify.

The premise for this experiment was to determine whether GSH depletion induces bacteriolysis, which we have previously reported will activate the inflammasome, and in the absence of caspases 1 and 11, will elicit cGAS-dependent IFN-I (Burke et al., *Nat Micro* 2019). Therefore, we hypothesized that if BSO was causing increased bacteriolysis, there would be more IFN-I. This was not the case, as reported in Fig 4D. The reviewer is asking why there is less IFN-I, which we agree could be due to fewer bacteria, however understanding this is complicated because there are already fewer bacteria in *Casp1/11* cells due to the increase in IFN-I (Burke et al., 2019). The reviewer could be correct that this is due to fewer bacteria, however that addresses a separate question unrelated to our hypothesis and we believe is tangential to the main premise of this experiment, which suggests that BSO does not cause bacteriolysis.

- line 209: missing the word 'factor' or 'regulator' after transcription

We fixed this error.

- the text could be more descriptive throughout. For example, lines 145-150 describe that lysis of *R. parkeri* leads to pyroptosis and then LDH is measured. It is assumed that the reader understands that pyroptosis leads to LDH release, which is what is measured as a proxy for pyroptosis. This should be clearly stated. Similarly for RLUs as a measurement of IFN release.

We have increased descriptions of the work throughout the manuscript, explicitly stating for example "Supernatants from infected cells were collected and cell death was measured using a lactate dehydrogenase (LDH) release assay, while IFN-I was used to stimulate an interferon-responsive cell line that produces luciferase in response to IFN-I. Relative Light Units (RLUs) were

then measured as a proxy for IFN-I.”

Reviewer #2 (Remarks to the Author):

In their manuscript, Sun et al describe the phenotypic outcomes of depletion of glutathione on intracellular replication of the obligate intracellular human pathogen, *Rickettsia parkeri*. *Rickettsia* species have undergone reductive evolution, leaving them with streamlined genomes and a reliance on a host cell for numerous essential metabolites. Glutathione has been implicated in virulence of a number of pathogens, motivating the authors to examine the role of glutathione in growth and pathogenesis of *R. parkeri*. Treatment of host cells with buthionine sulfoximine (BSO) to deplete glutathione during *R. parkeri* infection caused a number of defects, including reduced plaque size/number, cell division defects, reduced actin-based motility, and an inability to avoid autophagy. The authors propose that glutathione is required to support *R. parkeri* intracellular growth, perhaps through its role in redox homeostasis.

Overall this manuscript describes novel connections between a host metabolite and *R. parkeri* biology, although mechanisms linking glutathione to the phenotypes observed are not yet clear. Except for a few places noted below, the conclusions are largely supported by the data and the findings should be of interest to those interested in *Rickettsia* biology and host-pathogen interactions. The authors may want to consider the following suggestions to strengthen their manuscript:

1. The perturbation used to impact glutathione throughout the study is treatment with BSO. Does BSO impact host cell viability or replication?

To address any potential toxicity in our system, we have now performed a resazurin-based cell death assay and found no increase in cell death (Fig. S1). We also now include images of the BSO-treated and -untreated cells, where we observed no morphological changes in the cells or any reduction in cell number (Fig. S1). We also now better discuss the potential toxicity from BSO in different cell culture systems (lines ~80-90).

2. The authors report a “septation” defect with BSO treatment, which they define as cell chaining. The description and quantitation of this phenotype could be improved as follows:

a. The authors suggest that glutathione depletion causes bacterial cell chaining, which would be a cell separation defect, not a septation defect. Septation refers to formation of a septum, which the authors suggest does occur in BSO-treated *R. parkeri*.

We thank the reviewer for making us aware of this distinction. We have altered the text throughout and now refer to the defect as cell division or cell separation and not septation.

b. It is not clear from the images provided whether the cell division defect is actually in cell separation or in the formation of a septum (or both). Some cells do look chained, but others look more purely filamentous (without septa), in Figure 1C for example. Use of a cytoplasmic marker might be a more direct way to assess if the cytoplasm is compartmentalized in the filamentous cells. Alternatively, or in addition, electron microscopy would reveal the extent to which cells have (or do not have) a septum. Understanding which stage(s) of cell division is impacted would help inform hypotheses for how glutathione affects cell division.

We agree that many bacteria appear filamentous from the antibody-based staining. To address this, we performed the suggested experiment using GFP-expressing bacteria and super-resolution

microscopy. We observed in all cases that the bacteria have distinct fluorescent signals from the individual bacteria, and none appear to have a shared cytosol (Fig. 2D). We also note that with the phalloidin staining, some bacteria that appear filamentous from antibody-based staining have actin nucleation within the chain at sites approximately 1 μm apart, suggesting the formation of septums in the chained bacteria. This strengthens the conclusions that the bacteria have cell division defects and likely peptidoglycan separation defects and not a shared cytosol. We have better described this in the text.

c. From the images in Figure 1B, the cell division defect appears to be at its worst at ~48 hpi, with cells becoming shorter or more heterogeneous in length at 72 and 96 hpi. Is this the case? Can the authors quantify cell length as a function of time post infection across the population? It could be that cells are acquiring suppressors or otherwise overcoming the effects of glutathione depletion over time. Related to this, the authors report approximate cell lengths in the text - these analyses would be strengthened by reporting cell length for populations of treated and untreated cells. We have quantified and now report the cell length over time across the population. The bacteria increase in length over time from 24 – 72 hpi. Regarding suppressor mutations, we previously found that *R. parkeri* can accumulate mutations, but this requires serial passaging of >5 times over >20 days (Engström et al., 2019), and thus we would believe it to be unlikely for populations to gain suppressor mutations within 48 h. In this previous work we also have sequenced genomes from *R. parkeri* passaged many times in Vero cells and found few/no suppressor mutations, suggesting that they are similar to other Gram-negative bacteria and do rapidly not acquire mutations.

3. Lines 112-114. The effects of glutathione depletion on actin-based motility seem indirect, given the extended treatment time required to observe an effect. Do the authors think this is an indirect effect of the cell division defects they observe?

The reviewer is correct in that the consequences of GSH depletion on actin-based motility could be indirect. We observed that the bacteria undergo actin-based motility in the presence of BSO, but the tails are increasingly absent over time. We better discuss these possibilities in the Discussion and note that further experiments are required to address any potential direct or indirect role of GSH on actin-based motility.

4. Lines 123-128. The authors indicate that a putative glutathione S-transferase is encoded in Rickettsia species. The protein/gene mentioned actually appears to be a homolog of a cell division protein called FzIA that has been well-described in *Caulobacter crescentus* and, to a lesser extent, *Agrobacterium tumefaciens* (Goley et al 2010 Mol Cell, Lariviere et al 2018 Mol Micro, Lariviere et al 2019 Curr Biol, Mahone et al 2024 JCB). The *Caulobacter* FzIA homolog adopts a GST fold, but apparently does not bind glutathione in vitro - however, it seems possible that the *R. parkeri* protein might? If so, this could be a potential link between glutathione and cell division that should be mentioned, if not tested. Moreover, the transposon insertion in the gene encoding the *R. parkeri* FzIA homolog would disrupt the extreme C-terminus of the protein, so it may not be a complete loss of function. It is also possible that there are suppressing mutations elsewhere in the genome. We thank the reviewer for pointing us to these works, which we now reference in the Discussion. Based on the new data that NAC can rescue the GSH depletion phenotypes of chaining and survival, this suggests that *R. parkeri* is using cysteine as a nutrient source and thus unlikely that GstA is the link connecting GSH to cell division. Nevertheless the survival phenotype is not fully rescued so GSH could still play a role in division for example with the GstA protein. We now elaborate on this in the Discussion.

5. Lines 24 and 70-71: The authors conclude that their data support a paradigm of pathogens sensing host redox state as a cue for virulence, but this is not directly addressed experimentally. Given the other potential effects of glutathione on metabolism or protein function, etc, the authors might temper their conclusions to this effect. In addition, the effects reported here seem less effects on virulence, per se, and more on bacterial physiology (as the authors note in the discussion, lines 193-195).

We agree with the reviewer and have altered the text throughout the manuscript to reflect the role for GSH on cell division and physiology, including in the title, abstract, and elsewhere.

6. Line 31, “similar genome to *R. parkeri*” should be “similar genome to *R. rickettsii*”

We made this change.

Reviewer #3 (Remarks to the Author):

Significance

The manuscript entitled “Host glutathione is required for *Rickettsia parkeri* to properly septate, avoid ubiquitylation, and survive in macrophages” tries to establish a link between host cell glutathione and how it modulates *Rickettsia parkeri* intracellular growth and survival. Overall, the manuscript is well written and fairly easy to follow. The experimental approaches are based on published data that is both from the author’s group and other investigators. However, one of the main weaknesses of this work is that it provides a series of interesting observations, but does not address the mechanism(s) as to how *Rickettsia parkeri* utilizes host glutathione for the intracellular growth and dissemination. Specifically: The authors need to show how (or if) *R. parkeri* imports glutathione from the host. What is the mechanism of glutathione utilization by rickettsial metabolic pathways to support obligate intracellular life?

We thank the reviewer for their comments. In the revised manuscript we have made a significant advance in understanding the mechanism by investigating how GSH promotes *R. parkeri* cell division and intracellular growth. In the revised manuscript we have asked whether GSH is used as a nutrient source, as a post-translational modification, or for evading damage by reactive oxygen species. As described above, the additional experiments demonstrate that *R. parkeri* likely uses GSH as a cysteine source, which makes a significant advance in understanding the mechanism. We note that as an obligate cytosolic pathogen, understanding the role for individual metabolites is critical for understanding the intracellular lifecycle, yet is technically very challenging due to a lack of an axenic media to investigate mechanisms of nutrient uptake. Future studies will focus on better defining this connection between GSH uptake and cell division.

The authors just presented phenotypic data of rickettsial survival, septation, actin-based motility, ubiquitylation, autophagy in depleting glutathione using buthionine sulfoximine (BSO). No experimental evidence of glutathione levels in the host cells (uninfected, infected or with BSO treatment) was presented. This is surprising as 1) it is well known that BSO treatment modulates such levels and 2) would have helped to strengthen the significance of the authors findings. What are the endogenous glutathione levels of the utilized cells and are they different among the cell types (epithelial, endothelial and macrophages)? Could such potential differences in endogenous glutathione levels help explaining the phenotypical differences observed in these cell types? Experiments to measure GSH concentrations were not performed because these experiments have been extensively performed in previous work in the field, where it has been found that BSO

universally depletes GSH to <5% in a variety of cell types. We now better cite these previous reports in the text. Moreover, we note that these experiments are not easy to perform as they require a significant expertise in liquid chromatography. The reviewer is suggesting that perhaps GSH is more strongly depleted in BMDMs, which would account for the higher restriction in these cells. However, we note two important points that make this unlikely: 1) The bacteria in Vero cells are highly chained and growth is restricted, suggesting that GSH is strongly depleted. And 2) previous reports show that primary BMDMs restrict ubiquitylated *R. parkeri*, while cell lines including Vero cells do not (Engstrom *et al.*, 2019), and thus BMDMs are better at killing *R. parkeri* via antibacterial autophagy than cell lines.

The authors should provide data to why the authors chose the time and concentration of BSO utilized in the paper.

We chose these concentrations based on previous studies, for example 2 mM BSO was used to investigate the role for GSH in *Listeria monocytogenes* pathogenesis (Reniere *et al*, *Nature*, 2015) and at 500 uM to study the effects of GSH on *Burkholderia pseudomallei* pathogenesis (Wong *et al*, *Cell Host & Microbe* 2015). To address any potential concern about toxicity, we now include a new figure (Fig. S1) in which we measured toxic effects of BSO on host cells using a resazurin-based cell death assay, where we observed no increase in toxicity. We also now include images of untreated and BSO-treated cells 48 h after BSO treatment, which reveals that the cells maintain their normal morphology with no observable toxicity (Fig. S1).

The authors attempt to show a connection between glutathione metabolism and *R. parkeri* survival requires more rigorous examination. The authors should provide additional insights into how BSO treatment modulates other host cytokine responses (currently only focusing on IFN-I response) and how *R. parkeri* infection (+/- BSO treatment) modulates ROS production.

We thank the reviewer for this comment and now include additional data on how BSO affects other cytokine responses. We used a NF-kB reporter assay and found that *R. parkeri* infection of BSO-treated RAW cells elicited similar amounts of NF-kB activation as infection of untreated cells (Fig. 4D). This aligns with the data on inflammasome activation and IFN-I production, which suggests that the bacteria are being restricted via a non-lytic mechanism.

To address the reviewer's comment on how *R. parkeri* infection affects ROS, we have measured ROS abundance during infection, revealing that ROS is slightly decreased upon infection (Fig. 6C). BSO treatment slightly increases ROS production. We examined whether increasing ROS with hydrogen peroxide affects bacterial fitness and found that it neither affects bacterial abundance in BMDMs or chaining in Vero cells. These data contribute to our understanding of how GSH is used to promote cell division and suggest that ROS production is likely not responsible for the increased chaining and restriction.

Also, it is unknown whether the presented results (Figure 4) are related to Atg5-dependent autophagy or only ATG5 itself. Especially because ATG5-mediated host events independent of autophagy has been published previously.

We thank the reviewer for this idea and to address this concern we have now performed experiments in which we treated BMDMs with BSO and then an autophagy inhibitors (3MA). Interestingly, treatment of cells with this inhibitor restored a similar amount of *R. parkeri* survival (34x) as *Atg5* deletion (44x), suggesting that autophagy plays a significant role in restriction of *R. parkeri* upon GSH depletion.

The authors should further address the mechanism of how rickettsial molecules/effectors become

engaged in utilizing host glutathione to support intracellular growth and dissemination. We believe that this question is largely addressed in new data on GSH being required for a cysteine source to promote intracellular growth. Future studies for example on GSH transporters will further reveal this mechanism of GSH uptake and utilization. We also note that the *Rickettsia* field is not as advanced as that of many facultative cytosolic pathogens (ie *Listeria*, *Shigella*, *Burkholderia*) or other intracellular Gram-negative bacteria like *Legionella*, and we have a limited understanding of *Rickettsia* effectors altogether. This is due to the obligate cytosolic nature of these bacteria, which has limited the genetic toolbox, and generally slows progress in the field.

Response to reviewers (round 2, after accepted in principle).

Reviewer #1 (Remarks to the Author):

The authors have made significant progress and satisfactorily addressed all comments from the previous submission. The inclusion of new data strongly supports the conclusion that GSH serves as the primary source of sulfur/cysteine for cytoplasmic Rickettsia. However, the revised order of data presentation is somewhat confusing, and the interpretation of some data could be clearer. I recommend reorganizing the manuscript to integrate the new data into a more cohesive narrative. For example, Figures 3A-D could be combined with Figure 1A, as they all pertain to cell-cell spread. Similarly, replication and morphology defects could be consolidated into a single (or 2) figure(s) for clarity. Of course, the authors are free to structure the manuscript as they see fit, but the current arrangement reads more as a series of phenotypes rather than emphasizing the mechanistic insights. This approach obscures the depth of their work, which deserves to be highlighted.

We agree with the reviewer and we made these suggested changes to improve clarity, including moving the suggest panels around. We combined the previous figures 1 and 2 and moved the plaque assay (previous Fig 1A) to figure 2, which now emphasizes bacterial actin-based motility and cell-to-cell spread. We also moved 2 figures from the supplemental to the main Fig 1, as this will help with the flow and clarity of these sections.

In addition, using more precise language would strengthen the manuscript. For instance, the statement on line 244, “this is the first report linking the uptake of eukaryotic GSH to bacterial cell division,” implies that GSH is specifically linked to cell division. A more accurate phrasing would emphasize that GSH is required for bacterial replication. Improving language clarity will help to better convey the central conclusion: GSH serves as an essential nutrient for Rickettsia.

We removed the statement in question to avoid confusion for the reader and we made these suggested changes to improve clarity throughout the manuscript.

Specific Comments

Fig 1C does not appear to me to be a quantification of B. The BSO-treated line in C does not increase at all over 96 hours, suggesting ~ 2 bacteria per cell. Conversely, the images in C show a clear increase in the number of bacteria over time and far more than 2 bacteria in each cell shown.

We thank the reviewer for noticing this and we agree that the BSO-treated line in C did not increase, which was not aligned with the representative images. In the new Fig. 1J, we have increased the number of images analyzed and now show the individual data points on the graph, which shows a mild increase in bacterial numbers, better aligning with the representative images.

Importantly, 1C and 1D were not mentioned in the text, but these data clearly show a replication defect, which could explain the plaque defect. But the text ignores those results and instead investigates actin tails as an explanation for the plaque defect.

We have made sure to better emphasize the importance of each panel, and we agree that the plaque defect is both a combination of the chaining phenotype plus the actin-based motility phenotype. We now emphasize that the chaining is what likely causes the defects in actin-based motility, saying in line 120 that “We hypothesized that the aberrant morphology of *R. parkeri* upon GSH depletion would alter the ability of the bacteria to undergo proper actin-based motility.”

Fig 2: the addition of Fig 2D is really nice
Line 124 should call out Fig 3B

Line 127 should call out Fig 3D

We fixed the text so that each panel is mentioned and in order.

Fig 6: the images in B are not very informative as they are too small and zoomed out to see bacterial morphology. The quantification is sufficient so I would suggest eliminating the images to allow panels C and D to be bigger and more comprehensible.

We increased the size of the images by zooming in so that it is easier for the reader to appreciate the differences in each scenario.

Reviewer #2 (Remarks to the Author):

In their revised article, Sun and colleagues have provided additional data supporting a role of host glutathione (GSH) in promoting growth and division of *Rickettsia parkeri* as a source of cysteine. The demonstrate that treating infected cells with an inhibitor of host glutathione synthesis causes defects in growth, cell division, and autophagy evasion. Supplying cells with alternative cysteine sources rescues these defects. Overall the authors have addressed my major concerns from their initial submission. The inclusion of data to support a role of glutathione as a nutrient source helps to provide mechanistic insight into why depletion of GSH is deleterious. I have minor suggestions for improvement of the clarity of the manuscript.

1. Line 17 and 70. Instead of “chaining and survival were restored”, it should be “cell division and survival were restored”.

We agree and we made these suggested changes to indicate that cell division and survival were restored.

2. Lines 71-73. “...critical role for host GSH in *Rickettsia* cell division pathogenesis and contribute...” should be edited for clarity. Do you mean cell division and pathogenesis?

We altered this text for clarity, this was previously a typo.

3. Line 87. Change “...if BSO caused toxicity in our culture system...” to “if BSO caused toxicity to host cells in our system...” for clarity.

We made this change to note that BSO may cause toxicity to the cells.

4. Line 96. The reference to Fig 1B,C seems like it should be to Fig. 2B,C since that illustrates cell chaining. In general, it would be preferred to arrange figures and figure panels in the order to which they are referred in the text.

As mentioned above, we rearranged Figures 1-3 by combining Figs 1 and 2 and moving 1A to the new Figure 2. This improves clarity. We also made sure that figures were mentioned in the text in order.

5. Line 99. Add reference to Fig 1D, 2C.

We made these suggested changes (see above).

6. Line 122. “nucleated actin” would be better as “filamentous actin” since phalloidin specifically reports on F-actin.

We agree and we these suggested changes to filamentous actin.

7. Line 127. The image of HMEC cells shows *R. parkeri* with clouds... did any bacteria have tails in HMEC cells?

We now state more specifically that “Similar phenotypes were observed between HMEC-1 cells and Vero cells, including defects in cell division and actin-based motility” in line 132.

8. Line 189. Did the authors test if addition of GSH itself restored survival? That data was not shown, or I could not find it. If it does not, do the authors have an explanation for that? Can host cells not metabolize GSH to GSHee for import? Similarly, if cystine is the form of cysteine normally imported to Rickettsia can the authors explain why added cystine does not restore division and growth? That would seem to call into question the use of GSH as a cysteine source.

We had previously omitted GSH itself because we had explained that GSH is not imported, but instead GSHee is imported, which we found did restore division and growth. We now include GSH itself as Supplemental Fig. 1, as requested. We include references showing that GSH and other cysteine containing molecules including cystine are challenging for host cells to import, but GSHee or NAC are indeed imported, and these both restore *R. parkeri* division and growth. We add in additional explanation regarding how NAC is a widely used antioxidant because of its ability to be imported, thus NAC is a much more trustworthy cysteine source to use for these experiments than cystine.

9. For figure 6, consider re-ordering the panels so they are presented in the order they are described in the text.

We made these suggested changes by re-ordering and ensuring that they are presented as in the text.

10. Line 308. “Sca2 mediated” should have a hyphen.

We made these suggested changes to add a hyphen.

11. Fig 1 and legend, add what is labeled/visualized in panel B to the legend and, if possible, the image. (What is green, what is red?)

We have added additional details throughout the manuscript with the images to ensure clarity of the figures and coloring of images.

12. Figs 2, 3, 4, 6, S2. Again it would be helpful for clarity to indicate on the images what each panels is showing (actin, bacteria, etc).

We made these suggested changes, as described above.

13. The legends for Fig 3 and 4 also need a description of what is labeled. Also, the order of panels in Fig 3A is different than in B and D (i.e. green is shown first in A, 2nd in B and D). Consider making them consistent throughout the figure.

We made these suggested changes, as described above.

Reviewer #4 (Remarks to the Author):

In the revised manuscript, the authors presented additional data from new experiments supporting the role of GSH as a cysteine nutrient source. All concerns raised by this reviewer are addressed.

We thank the reviewer for their time and insight.

There are two minor typo's that need to be fixed.

Line 306: Should read as " cell division, which then...."

We made these suggested changes.

Line 309: Should read "surfaces later in the infection."

We made these suggested changes.